# lncRNA *CARINH* regulates expression and function of innate immune transcription factor IRF1 in macrophages

Yannick Cyr[1], Morgane Gourvest[1], Grace O Ciabattoni[2], Tracy Zhang[1], Alexandra AC Newman[1], Tarik Zahr[1], Sofie Delbare[1], Florencia Schlamp[1], Meike Dittmann[2], Kathryn J Moore[1,3], Coen van Solingen[1]

**The discovery of long non-coding RNAs (lncRNAs) has provided a new perspective on the centrality of RNA in gene regulation and genome organization. Here, we screened for lncRNAs with putative functions in the host response to single-stranded RNA respiratory viruses. We identify *CARINH* as a conserved cis-acting lncRNA up-regulated in three respiratory diseases to control the expression of its antisense gene *IRF1*, a key transcriptional regulator of the antiviral response. *CARINH* and *IRF1* are coordinately increased in the circulation of patients infected with human metapneumovirus, influenza A virus, or SARS-CoV-2, and in macrophages in response to viral infection or TLR3 agonist treatment. Targeted depletion of *CARINH* or its mouse ortholog *Carinh* in macrophages reduces the expression of IRF1/ Irf1 and their associated target gene networks, increasing susceptibility to viral infection. Accordingly, CRISPR-mediated deletion of *Carinh* in mice reduces antiviral immunity, increasing viral burden upon sublethal challenge with influenza A virus. Together, these findings identify a conserved role of lncRNA *CARINH* in coordinating interferon-stimulated genes and antiviral immune responses.**

## Introduction

Human respiratory viruses, including severe acute respiratory syndrome coronavirus 2 (SARS-CoV-2), metapneumovirus (MPV), and influenza A virus (IAV), pose a significant threat to global health (Jackson et al, 2021; Tsalik et al, 2021). Effective antiviral immunity relies on the production of type I interferons (IFNα/β) and the coordinated expression of hundreds of IFN-stimulated genes (ISGs) with antiviral and immunomodulatory functions (Trinchieri, 2010; Gonzalez-Navajas et al, 2012; Kopitar-Jerala, 2017). Type I IFNs are secreted proteins that act locally and systemically by engaging the type I IFN receptor (IFNAR) and triggering Janus kinase/signal transducers and activators of transcription (JAK-STAT) signaling.

This results in assembly of the ISG factor 3 complex, consisting of a STAT1/STAT2 heterodimer and IFN regulatory factor 9 (IRF9), that translocates to the nucleus where it binds to IFN-stimulated response elements (ISRE) in the promoter region of target genes (Andres-Terre et al, 2015; Zhai et al, 2015; Tsalik et al, 2021). This inflammatory cascade is further propagated by the expression of additional IFN regulatory factors (IRFs), some of which are themselves ISGs. IRFs act downstream of the JAK-STAT pathway by inducing the transcription of IFNs, ISGs, and additional antiviral effector genes (Panda et al, 2019; Wang et al, 2020; Zhou et al, 2022). In addition, IRF1 has been shown to modulate phosphorylation and localization of IRF3/7 and JAK-STAT, thereby fine-tuning IFN and other proinflammatory responses. Notably, the antiviral response needs to be carefully controlled in magnitude, timing, and location to avoid overt tissue damage, including acute respiratory distress or cytokine release syndromes (Rouse & Sehrawat, 2010; Shi et al, 2020).

Long non-coding RNAs (lncRNAs) are increasingly recognized as an important layer of gene regulation within immune response pathways (Atianand et al, 2016; Vierbuchen & Fitzgerald, 2021; van Solingen et al, 2022). lncRNAs act by numerous mechanisms including forming ribonucleoprotein complexes that can function as guides or decoys, shaping of nuclear organization and higher order chromosomal architecture (Goff & Rinn, 2015; Satpathy & Chang, 2015), and scaffolding of RNA or protein effector partners to attenuate or enhance regulatory activities (Wang & Chang, 2011). lncRNAs can function in cis, close to their site of transcription to regulate the expression of neighboring genes, or in trans, at distal sites in the genome (Guil & Esteller, 2012; Joung et al, 2017). This class of heterogeneous transcripts, arbitrarily defined as being >200 nucleotides in length, exhibits low abundance and poor conservation among species—factors that have hindered their investigation (Palazzo & Lee, 2015; Schmitz et al, 2016). Despite the abundance of lncRNAs in the human transcriptome, fewer than 30 lncRNAs have been mechanistically described in response to viral infection (Kesheh et al, 2022).

Numerous neighboring lncRNA and mRNA pairs located within a topologically associated domain (TAD) have been described to be

[1]Department of Medicine, Cardiovascular Research Center, New York University Grossman School of Medicine, New York, NY, USA [2]Department of Microbiology, New York University Langone Health, New York, NY, USA [3]Department of Cell Biology, New York University Langone Health, New York, NY, USA

Correspondence: coen.vansolingen@nyulangone.org

transcriptionally co-regulated (Khyzha et al, 2019), suggesting that potential shared functions may be inferred from the known role of the protein-coding gene. A prime example induced by viral infection is the co-transcription of the lncRNA BST2 Interferon Stimulated Positive Regulator (*BISPR*, *lncBST2*) and its proximal gene Bone Marrow Stromal Cell Antigen 2 (*BST2*, tetherin), which encodes a protein that prevents the detachment of enveloped virus particles from infected cells. *BISPR* and *BST2* share a bidirectional promoter, and both transcripts are up-regulated upon stimulation with IFN or after infection with vesicular stomatitis virus. Knockdown of *BISPR* using siRNA in human hepatocarcinoma cells (Huh-7) or alveolar basal epithelial cells (A549) leads to the down-regulation of *BST2*, but not other neighboring genes within the TAD, suggesting a specific role of *BISPR* in the regulation of tetherin activity (Barriocanal et al, 2014). Further characterization of lncRNAs that regulate antiviral immune responses has the potential to reveal novel layers of regulation and potential therapeutic targets.

lncRNA Colitis Associated IRF1 antisense Regulator of INtestinal Homeostasis (*CARINH*) (Ma et al, 2023), also known as *C5ORF56* (Chiaroni-Clarke et al, 2014) and *ISR8/IRF1-AS1* (Barriocanal et al, 2022), is located on the opposite strand of the interferon regulatory factor 1 (*IRF1*) gene, in the antisense direction. *CARINH* has been shown to be induced by IFN in esophageal squamous cell carcinoma and HeLa cells (Huang et al, 2019; Barriocanal et al, 2022), and deletion of its promoter region leads to decreased cell survival upon infection with encephalomyocarditis virus (Barriocanal et al, 2022). Yet, the molecular mechanisms through which the *CARINH* transcript regulates the IFN response remain unclear. In this study, we show that *CARINH* is highly expressed in the circulation of patients infected with MPV, IAV, or SARS-CoV-2, and is induced in macrophages exposed to IAV, synthetic viral mimic dsRNA polyinosinic–polycytidylic acid (poly[I:C]), or IFNβ. Loss-of-function studies identify a critical role of *CARINH* in the regulation of IRF1 and downstream ISG expression, and, consequently, restriction of IAV replication in macrophages. Synteny analysis of the human and mouse genomes revealed that *CARINH* is among the minority of human lncRNAs with a mouse ortholog, *Carinh* (also *Gm12216*), which is located antisense to mouse *Irf1*. Knockdown studies of *Carinh* showed a down-regulation in ISG transcription, reproducing observations made for *CARINH*. Furthermore, CRISPR/Cas9-engineered *Carinh*⁻/⁻ mice challenged with IAV present reduced inflammatory symptoms and, consequently, increased short-term survival. Collectively, our data provide insight into the role of *CARINH* and its murine ortholog *Carinh* in regulating the IFN transcriptional program upon viral infection.

## Results

### CARINH and its proximal gene IRF1 are co-induced upon viral infection

To identify lncRNAs with putative functions in the host response to single-stranded RNA respiratory viruses, we compared whole blood transcriptomic analyses of patients presenting an infection with human metapneumovirus (MPV, n = 8) or influenza A virus (IAV,

n = 41), with age- and sex-matched controls (n = 18, GSE157240 [Tsalik et al, 2021]), or severe acute respiratory syndrome coronavirus 2 (CoV-2, n = 8; controls, n = 7; GSE190413 [van Solingen et al, 2022]) (Fig 1A). Differential expression analyses revealed that 282 lncRNAs were altered in individuals infected with MPV, 418 lncRNAs in those infected with IAV, and 813 in those infected with SARS-CoV-2, compared with their respective control populations (Fig 1B, *P*-adj < 0.05, Table S1). Furthermore, 137 lncRNAs were up-regulated in all three viral infections, suggesting fundamental roles of these lncRNAs in the host response to respiratory viral infection in humans (Fig 1C). Among the lncRNAs differentially expressed across all three diseases, we noted several lncRNAs previously reported to be involved in antiviral innate immune responses, including *BISPR* (Barriocanal et al, 2014), *CCR5AS* (Kulkarni et al, 2019), *CHROMR* (van Solingen et al, 2022), *NRIR* (Mariotti et al, 2019), and *CARINH* (Fig 1B, Table S1). Within our dataset, we also performed differential expression analysis to selectively capture protein-coding genes (Fig S1A) and found 3,811 mRNAs with altered expression across all three viral infections (Fig 1D).

To screen for *cis*-regulatory lncRNAs that may control the expression of proximal coding genes induced by viral infection, we implemented an unbiased approach to identify candidate lncRNA-mRNA pairs that are differentially expressed in viral infection compared with controls and localized within a genomic distance of 5 kb (Khyzha et al, 2019). This analysis identified 44 putative cis-regulatory lncRNA-mRNA pairs (Figs 1E and S1B, Table S2) that were significantly differentially expressed in all three diseases. To select candidates for further study, we rank-sum–ordered lncRNA-mRNA pairs based on the ranking of each lncRNA's differential up-regulation in the three infectious diseases. In whole blood of patients infected with MPV, IAV, or SARS-CoV-2, the top differentially expressed putative cis-regulatory lncRNA-mRNA pairs were G-quadruplex Forming Sequence Containing lncRNA (*GSEC*) and ST3 beta-galactosidase alpha-2,3-sialyltransferase 4 (*ST3GAL4*), *LINC02422* and Retroelement Silencing Factor 1 (*RESF1*), and Colitis Associated IRF1 antisense Regulator of Intestinal Homeostasis (*CARINH*) and Interferon Regulatory Factor 1 (*IRF1*) (Figs 1F and S1B).

Next, we interrogated the transcriptional co-regulation between each lncRNA and its proximal coding gene in patients infected with IAV (n = 41). Using a linear regression analysis, we observed a significant association between the expression of the lncRNA and its neighboring mRNA for 35 of the 44 identified pairs, suggesting shared transcriptional regulation (Fig 2A, Table S3). To determine whether these lncRNA-mRNA pairs are regulated in myeloid cells in response to IAV infection, we leveraged RNA-sequencing (RNA-seq) data from human monocyte-derived macrophages infected with A/California/04/09 (H1N1), influenza A/Wyoming/03/03 (H3N2), or influenza A/Vietnam/1203/2004 (H5N1 HaLo) viruses (retrieved from GSE97672 [Heinz et al, 2018]). Of our top candidate cis-acting lncRNAs, we found that *LINC02422* and *CARINH* were significantly up-regulated in macrophages upon infection with any of the influenza strains, whereas *GSEC* remained unresponsive (Fig 2B). By comparison, analysis of the expression of their proximal genes showed that only *IRF1*, but not *ST3GAL4* and *RESF1*, was increased by viral challenge in primary monocyte-derived macrophages (Fig 2C). Taken together, these data identify *CARINH* and its proximal coding gene *IRF1* as a putative cis-acting lncRNA-mRNA pair induced by viral infection in humans.

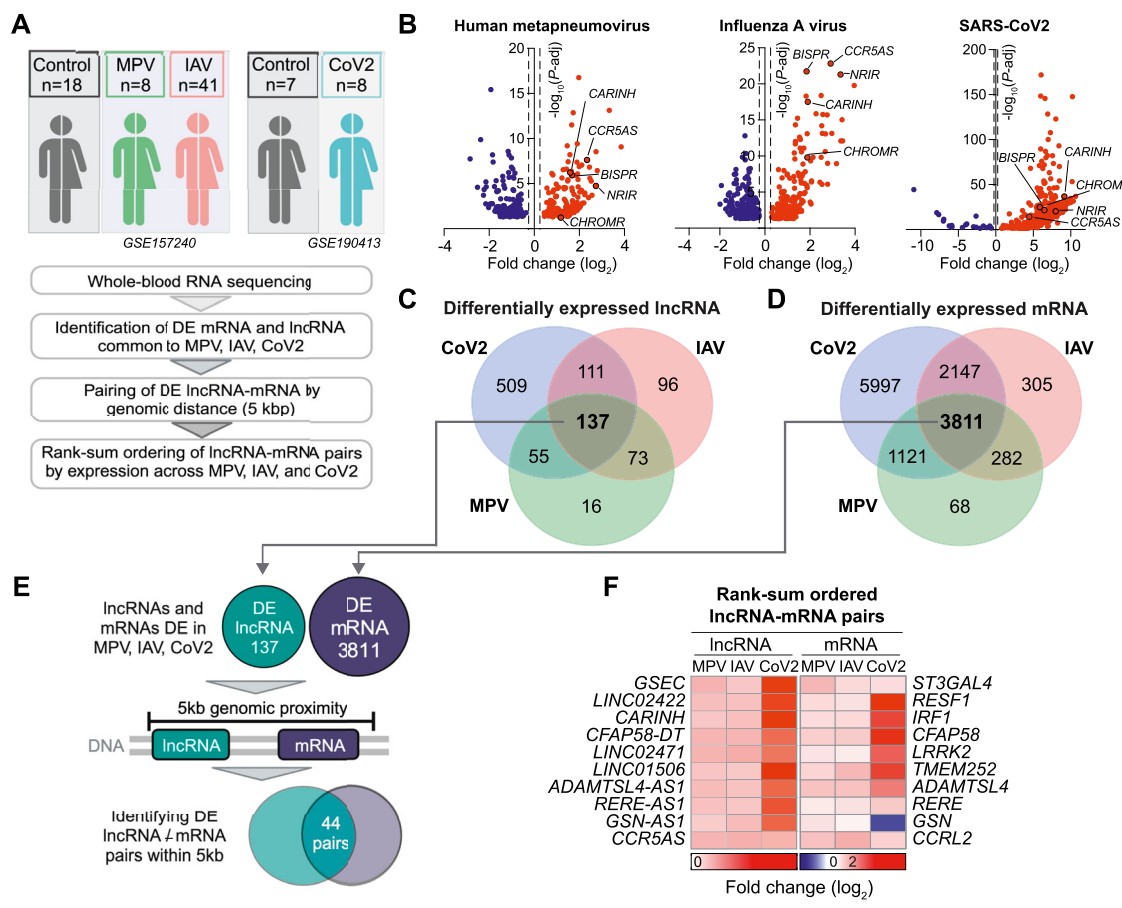

**Figure 1. Long non-coding RNA (lncRNA) and proximal coding mRNA pairs are up-regulated in patients infected with metapneumovirus, influenza A, and SARS-CoV-2.**
**(A)** Experimental approach used to identify lncRNA-mRNA pairs encoded within the same topologically associated domain (<5 kb) and differentially expressed in whole blood of patients with metapneumovirus (MPV), influenza A virus (IAV), or SARS-CoV-2 (CoV-2), and controls. **(B)** Volcano plots of differentially expressed lncRNA in whole blood of patients with MPV, IAV, or CoV-2 compared with controls. Dashed lines indicate fold change (log$_2$) = ±0.25; $P$-adj = 0.05. **(C, D)** Venn diagrams showing numbers of differentially expressed lncRNA (C) and mRNA (D) in whole blood of patients with MPV, IAV, or CoV-2 compared with controls. **(E)** Strategy used to identify lncRNA-mRNA pairs differentially expressed in MPV, IAV, or CoV-2 and within 5-kb genomic vicinity. **(F)** Heatmap of the rank-sum–ordered expression of top 10 lncRNA and proximal coding mRNA in MPV, IAV, or CoV-2.

## CARINH regulates ISG expression in response to pathogen sensing

The *CARINH* gene (ENSG00000197536.11) is located on human chromosome 5 and has three splice variants (ENST00000612967.2 [*CARINH-V1*], ENST00000337752.6 [*CARINH-V2*], and ENST00000407797.6 [*CARINH-V3*]), which share the first three exons but differ at their 5′ end. *CARINH* is positioned antisense to IRF1, with *CARINH-V1* overlapping the IRF1 coding sequence (Fig 3A). Using PCR primers directed at common sequences in exon 1, we found that *CARINH* and *IRF1* transcripts are concurrently up-regulated in human primary CD14⁺ monocyte-derived macrophages treated with IFNβ for 8 h (Fig 3B). To investigate the regulation of specific *CARINH* variants, we examined how the expression of *CARINH* variants was altered by stimulation with IFNβ or the synthetic Toll-like receptor (TLR)3 agonist polyinosinic–polycytidylic acid (poly[I:C]). In THP1 macrophages treated with IFNβ, we observed a time-dependent increase in expression levels of all three *CARINH* splice variants coincident with *IRF1* up-regulation that reached statistical significance after 24 h (Fig S2A). Similarly, in response to TLR3 stimulation

by poly(I:C), *CARINH-V1* and *IRF1* were significantly induced after 8 h, whereas *CARINH-V2* and *CARINH-V3* showed a similar trend (Fig S2B).

Given that their transcriptional orientation is divergent, the co-expression of *CARINH* and *IRF1* is unlikely to be driven by a common promoter or transcriptional activation event, suggesting in cis regulation by *CARINH*. To investigate the chromatin architecture of the topologically associated domain containing *CARINH* and *IRF1*, we visualized the three-dimensional (3D) architecture within this genetic locus using high-throughput chromosome conformation capture (Hi-C) (Lieberman-Aiden et al, 2009) data from the 3D genome browser (Wang et al, 2018) in THP1 macrophages (Phanstiel et al, 2017). Compared with other proximal genes (e.g., *RAD50* or *SLC22A5*), we observed enhanced chromatin interactions between the *IRF1* and *CARINH* loci, indicating putative formation of chromatin loops driven by *CARINH* (Fig 3C). Notably, the expression levels of other genes in this topologically associated domain did not correlate with *CARINH* in IAV-infected patients (Fig S2C and D), suggesting a specific contact between *CARINH* and the *IRF1* gene

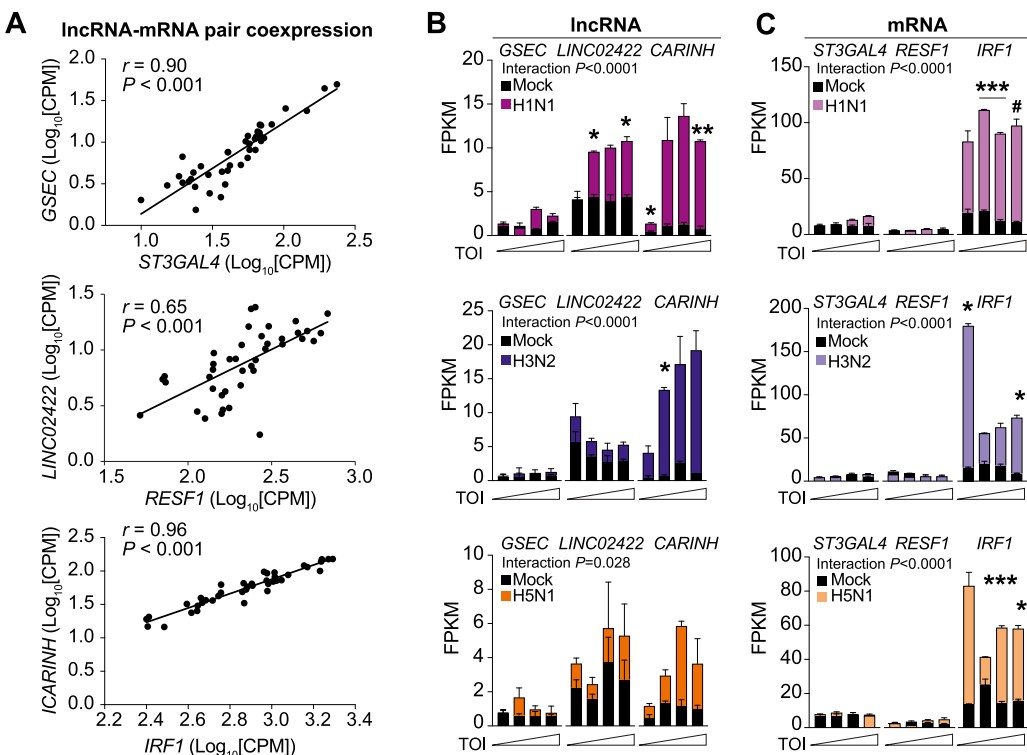

**Figure 2. Long non-coding RNA *CARINH* and its proximal coding gene *IRF1* are co-expressed in macrophages upon influenza A infection.**
**(A)** Pearson's correlation analysis and regression line of long non-coding RNA (*GSEC, LINC02422, CARINH*) and proximal coding gene (*ST3GAL4, RESF1, IRF1*, respectively) expression in whole blood of patients with influenza A virus (IAV, n = 41). **(B, C)** Time course of *GSEC, LINC02422, CARINH* (B), and *ST3GAL4, RESF1, IRF1* (C) expression (FPKM) in human monocyte-derived macrophages infected for 3, 6, 12, or 18 h (TOI: time of infection) with influenza A/California/04/09 (H1N1), influenza A/Wyoming/03/03 (H3N2), or influenza A/Vietnam/1203/2004 (H5N1), or mock-infected. Data are the mean ± SEM for two independent experiments. *P*-values were calculated by repeated-measures two-way ANOVA with Tukey's multiple-comparisons test between IAV infection and mock control (B, C). #*P* < 0.1; *P* < 0.05; **P* < 0.01; ***P* < 0.001.

locus. To further validate a potential interaction between *CARINH* and the *IRF1* gene locus, we performed Chromatin Isolation by RNA Purification (ChIRP) of endogenous *CARINH* in nuclear extracts of cross-linked THP1 macrophages using two independent pools of *CARINH*-specific antisense RNA probes (Pool 1, Pool 2) and LacZ controls. Isolation of *CARINH*-associated chromatin followed by DNA sequencing (ChIRP-seq) revealed enrichment of *CARINH* at 232 loci mostly within intronic and intergenic regions (Fig 3D). Within the regions that were most strongly enriched for *CARINH* was an intronic region within IRF1 and the transcriptional termination site of IL18BP, a gene previously shown to be controlled by *CARINH* (Ma et al, 2023) (Fig 3E and F).

To test whether depleting *CARINH* alters *IRF1* expression and downstream responses in macrophages, we transfected human macrophages with *CARINH*-targeting (Gap*CARINH*) or control (GapCTRL) GapmeR antisense oligonucleotides. Treatment with Gap*CARINH* decreased *CARINH* transcript levels in both primary CD14+ monocyte-derived macrophages and THP1 macrophages, as visualized by RNA FISH (Fig 4A) or qRT–PCR (Fig 4B). Notably, transfection with Gap*CARINH* also reduced *IRF1* transcript levels in both CD14+ monocyte-derived and THP1 macrophages when compared to non-targeting GapCTRL, as measured by qRT–PCR (Fig 4B). As IRF1 plays a central role in the type I IFN response, we next profiled the transcript levels of 84 selected ISGs in THP1 macrophages using a qRT–PCR array. Compared with GapCTRL, Gap*CARINH*

treatment significantly reduced the expression of more than 25% of ISGs measured including critical regulators of viral defense through viral RNA degradation (*OAS1, OAS2*), cytokine signaling (*IL6*), and inhibition of viral replication (*IFIT2, IFIT3*) (Fig 4C). In contrast, only one ISG, PRKCZ, was up-regulated upon Gap*CARINH* treatment (Fig 4C). Of note, 14 of the 23 differentially expressed ISGs show significant direct physical and functional interaction (Fig 4D), suggesting that *CARINH* contributes to the coordinated expression of ISGs required for antiviral immunity and the amplified production of IFNs. Accordingly, depletion of *CARINH* in macrophages treated with poly(I:C) blunted the secretion of IFNβ, IFNγ, and IFNλ protein, as measured by cytometric bead immunoassay (Fig 4E), indicating an important role in regulating IFN response. In addition, knockdown of *CARINH* in THP1 macrophages using Gap*CARINH* resulted in the down-regulation of IRF1 protein levels, compared with GapCTRL-treated macrophages (Fig 4F).

To test whether *CARINH* regulates ISG expression by altering IRF1 transcriptional activation of ISRE-bearing target genes, we used THP1 macrophages stably expressing an ISRE-inducible reporter construct. Treatment with Gap*CARINH* significantly reduced poly(I:C)- or IFNβ-induced ISRE-reporter expression compared with GapCTRL treatment (Fig 4G), whereas basal ISRE-reporter response levels were unaffected by transfection with Gap*CARINH* or GapCTRL (Fig S3A). As these data suggest that *CARINH* is a critical regulator of the ISG network induced during antiviral immunity, we next

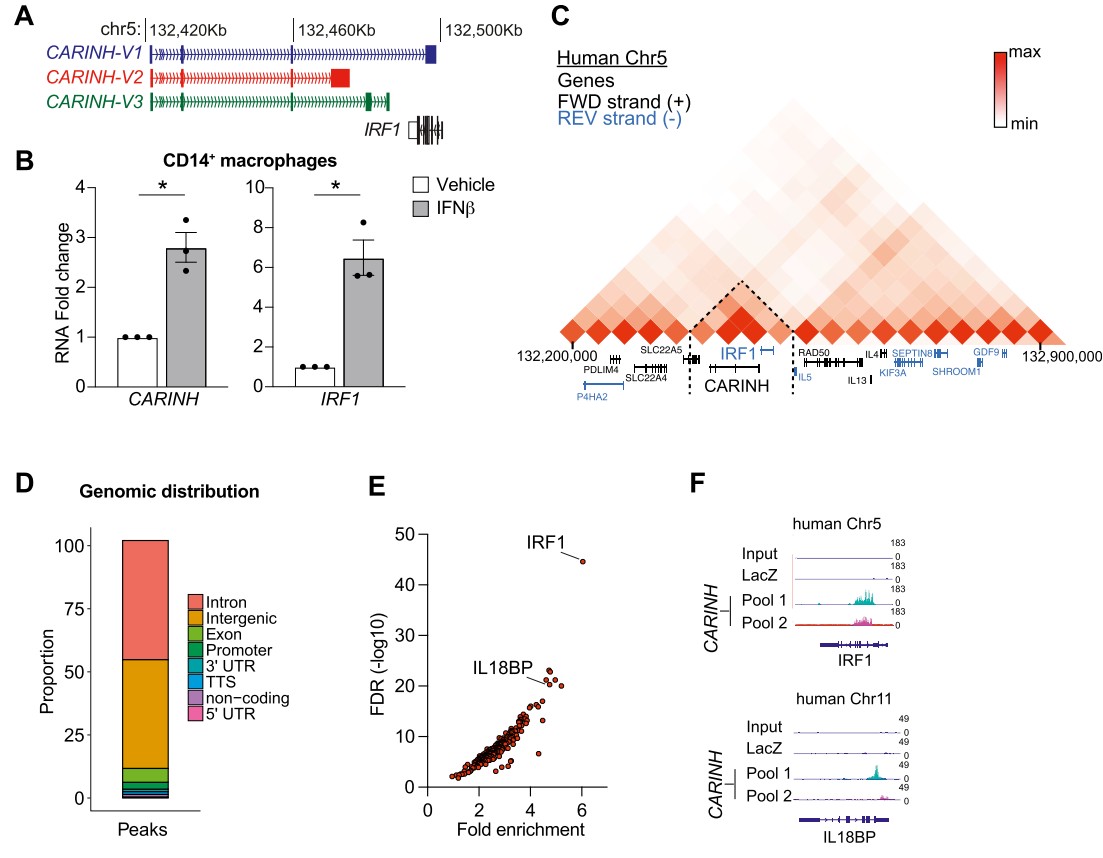

**Figure 3.** ***CARINH*** **is up-regulated after a type I IFN challenge and binds to the IRF1 gene locus.**
**(A)** Schematic representation of the human *CARINH* locus on chromosome 5, which encodes three splice variants (*CARINH-V1-3*) located antisense to IRF1. Solid boxes indicate exonic sequences. Arrows indicate the direction of transcription. **(B)** qRT–PCR analysis of *CARINH* and *IRF1* in CD14[+] primary human macrophages treated with 500 U/ml IFN*β* for 8 h. **(C)** Hi-C heatmap of chromatin interactions at the genomic location of *CARINH* in THP1 macrophages. Black writing indicates genes on the forward strand, and blue writing indicates genes on the reverse strand. Dashed lines indicate the area of interaction. **(D)** Distribution of *CARINH* binding sites within specific loci of the human genome. **(E)** Volcano plot showing enriched loci bound by *CARINH* (n = 232). **(F)** ChIRP-seq reads for IRF1, IL18BP. Top row: input; second row: LacZ control; third row: *CARINH* probe set 1; and bottom row: *CARINH* probe set 2. Data are the mean ± SEM for three independent experiments (B). *P*-values were calculated by a one-sample *t* and Wilcoxon test (B). *$P < 0.05$.

assessed the role of *CARINH* in restricting IAV replication in human macrophages. THP1 macrophages were treated with Gap*CARINH* or GapCTRL and challenged with influenza A/WSN/1933 (H1N1) virus at a multiplicity of infection of 1. We assessed the level of infection 24 h later by high-content microscopy quantification of cellular IAV nucleoprotein. Knockdown of *CARINH* significantly increased the percentage of IAV-infected macrophages compared with GapCTRL treatment, leading to a doubling of infection and to decreased cell viability (Fig 4H). Of note, this was not caused by enhanced apoptosis driven by *CARINH* depletion, as transfection of GapCTRL or Gap*CARINH* did not induce cell toxicity (Fig S3B). Together, our results suggest a role of *CARINH* in regulating *IRF1* expression and coordinating the expression of ISGs required to limit viral infection in human innate immune cells (Fig 4I).

## *Carinh* is a mouse ortholog of *CARINH* with conserved functions

Although many human lncRNAs are poorly conserved, we observed compelling similarities between the *IRF1* locus on human chromosome 5 and the *Irf1* locus on mouse chromosome 11, where

lncRNA *Gm12216* (*Carinh*) is positioned convergent to *Irf1* (Fig 5A). Synteny analysis showed considerable orthologous alignment between *CARINH* and *Carinh* suggesting conservation from human to mouse (Fig 5B). Furthermore, similar to human *CARINH*, we observed enhanced chromosomal interactions between the *Carinh* and *Irf1* loci but no other proximal genes by Hi-C analysis (Fig S3C). To determine whether *Carinh* is functionally related to *CARINH*, we first treated mouse bone marrow–derived macrophages (BMDM) with poly(I:C) or IFN*β* for 8 h. We observed concordant up-regulation of *Carinh* and *Irf1* transcript levels in response to these stimuli compared with vehicle control (Fig 5C and D). In addition, knockdown of *Carinh* in BMDM using GapmeR treatment (Gap*Carinh*) resulted in the down-regulation of both *Carinh* and *Irf1* transcript levels and IRF1 protein levels compared with GapCTRL-treated cells (Fig 5E and F). To assess the role of *Carinh* in IRF1-dependent transcriptional activation of ISGs, we transfected mouse RAW264.7 macrophages stably expressing an ISRE-inducible reporter with control or *Carinh*-targeting GapmeRs and treated with poly(I:C), IFN*β*, or a vehicle control. As we observed in *CARINH* loss-of-function studies in human THP1 cells (Figs 4G and S3A), depletion

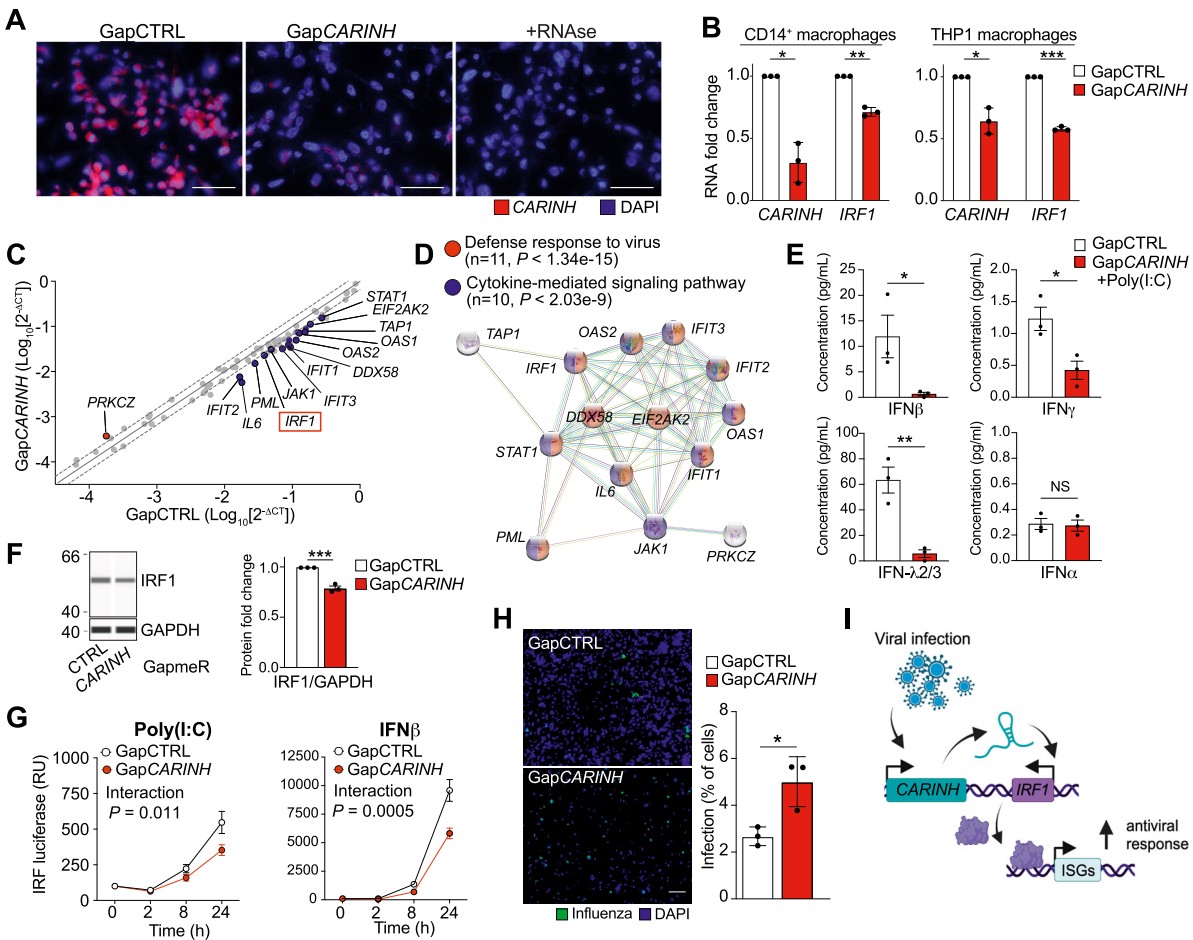

**Figure 4. Knockdown of *CARINH* leads to impaired IRF-driven immune response.**
**(A)** FISH of *CARINH* (red) RNA in THP1 macrophages treated with GapCTRL, or GapmeRs targeting *CARINH*, or an RNase control, and DNA counterstain (DAPI, blue). The scale bar is 50 μm. **(B)** qRT–PCR analysis of *CARINH* and *IRF1* in *CARINH*-depleted (Gap*CARINH*-treated) and control (GapCTRL-treated) CD14⁺ primary human macrophages (left) or THP1 macrophages (right). **(C)** qRT-PCR array–based gene expression profiling of 84 type I interferon response genes in THP1 macrophages treated with Gap*CARINH* versus GapCTRL. **(C, D)** Interactome of significant differentially expressed genes shown in (C). Red/blue colors indicate belonging to the indicated canonical pathway. **(E)** Cytometric bead array of IFNβ, IFNγ, IFNλ2/3, and IFNα protein levels in the supernatant of THP1 in Gap*CARINH* versus GapCTRL-treated cells, and subsequent treatment with poly(I:C) (1 μg/ml) for 24 h. **(F)** Western blot analysis of IRF1 in Gap*CARINH*-treated and GapCTRL-treated THP1 macrophages. **(G)** Reporter assay for IRF-driven transcription in human THP1-Lucia reporter macrophages transfected with Gap*CARINH* or GapCTRL and treated with or poly(I:C) or IFNβ (500 U/ml). Relative luciferase expression (relative units [RU]) is normalized to time 0 h, set at 100%. **(H)** Representative whole-well microscopy images of immunofluorescent staining for influenza A–infected (H1N1, green) THP1 macrophages transfected with Gap*CARINH* and GapCTRL counterstained for nuclear DNA (DAPI, blue). Quantification shown on right as a percentage of infected cells per total number of viable cells in *CARINH*-depleted and control THP1 macrophages. The scale bar is 500 μm. **(I)** Integrated model depicting control of *IRF1* expression by *CARINH* leading to alteration of the antiviral immune response. Data are the mean ± SEM for three independent experiments (A, E, F, G) or representative of three independent experiments. *P*-values were calculated by a one-sample *t* and Wilcoxon test (B, F), a *t* test (E, H), and repeated-measures two-way ANOVA with significant group differences between Gap*CARINH* and GapCTRL (G). \**P* < 0.05; \*\**P* < 0.01; \*\*\**P* < 0.001.

of *Carinh* led to diminished ISRE-dependent transcriptional activation in poly(I:C)- and IFNβ-treated macrophages, whereas no differences were found in untreated cells (Figs 5G and S3D). Together, these data provide further evidence that *Carinh* is an ortholog of *CARINH* with conserved functions in regulating macrophage innate immune responses through control of *Irf1* expression.

## Deletion of *Carinh* in mice impairs antiviral immunity

To study the function of *Carinh* in vivo, we used CRISPR/Cas9 technology to generate a *Carinh*-knockout mouse. Single-stranded oligo-deoxynucleotides containing the poly(A) sequence of SV40

were used to create an 82 base-pair indel (Yoshimi et al, 2016; Liu et al, 2017) in the first exon of *Carinh*, resulting in termination of the *Carinh* transcript (Fig 6A). The *Carinh* exon 1 indel was confirmed by PCR genotyping of homozygous indel-bearing mice (*Carinh*⁻/⁻), and RNA-seq of BMDM isolated from *Carinh*⁻/⁻ mice showed the absence of the *Carinh* transcript compared with wild-type (WT) littermates (Fig 6B and C). Loss of *Carinh* did not induce cellular cytotoxicity as shown by propidium iodide and annexin V staining of bone marrow cells (Fig S4A). Targeted transcriptomic analysis of RNA-seq data from WT and *Carinh*⁻/⁻ BMDMs treated with IFNβ, using an IRF1 gene signature predicted by the upstream transcriptional regulator analysis database of Ingenuity Pathway Analysis (QIAGEN), revealed a marked shift

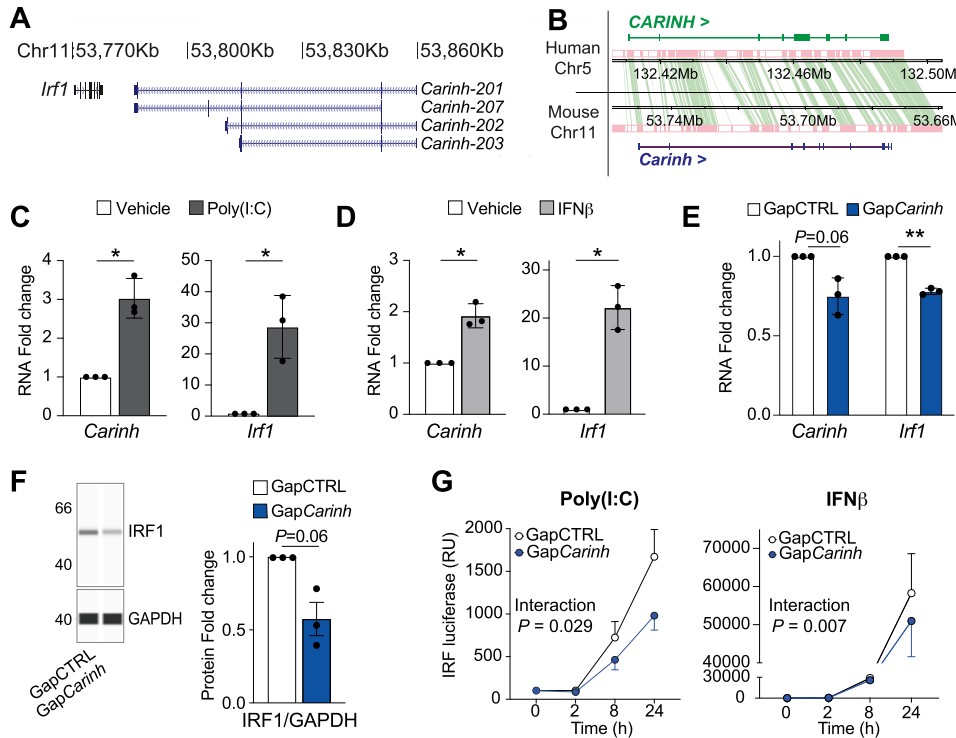

**Figure 5. *Carinh* is a mouse ortholog of *CARINH* with a similar function.**
**(A)** Schematic representation of the *Carinh* locus on mouse chromosome 11, which encodes four splice variants. Solid boxes indicate exonic sequences. Arrows indicate the direction of transcription. **(B)** Synteny analysis of the human *CARINH* and mouse *Carinh* loci showing regions of conservation in pink, connected by green bands. **(C, D)** qRT-PCR analysis of *Carinh* and *Irf1* in bone marrow–derived macrophages (BMDMs) after 8-h treatment with poly(I:C) (1 μg/ml, (C)) or IFNβ (1,000 U/ml, (D)) compared with vehicle control. **(E)** qRT–PCR analysis of *Carinh* and *Irf1* in *Carinh*-depleted (Gap*Carinh*-treated) and control (GapCTRL-treated) BMDMs. **(F)** Western blot analysis of IRF1 in Gap*Carinh*-treated and GapCTRL-treated BMDMs. **(G)** Reporter assay for IRF-driven transcription in mouse RAW-Lucia reporter macrophages transfected with Gap*Carinh* or GapCTRL and subsequently treated with poly(I:C) (1 μg/ml) or IFNβ (1,000 U/ml). Relative luciferase expression (relative units [RU]) is normalized to time 0 h, set at 100%. Data are the mean ± SEM for three independent experiments (C, D, E, F, G). *P*-values were calculated by a one-sample *t* and Wilcoxon test (C, D, E, F); or repeated-measures two-way ANOVA with significant group differences between Gap*Carinh* and GapCTRL (G). **P* < 0.05; ***P* < 0.01.

of the IRF1-driven gene signature in *Carinh*$^{-/-}$ BMDMs compared with similarly treated WT cells. *Carinh*$^{-/-}$ BMDMs showed the reduced expression of key innate immune genes like *Ptgs2*, *Cdh2*, *Mmp9*, and *Il6* in response to IFNβ, compared with WT BMDMs (Fig 6D). To assess whether the attenuated IRF1 gene signature was due to a reduction of IRF1 protein in *Carinh*$^{-/-}$ cells, we performed immunohistochemical staining for IRF1 protein in BMDMs and peritoneal macrophages from *Carinh*$^{WT}$ and *Carinh*$^{-/-}$ mice after treatment with IFNβ, poly(I:C), or vehicle. Indeed, macrophages derived from *Carinh*$^{-/-}$ mice showed lower levels of IRF1 under both treated and untreated conditions (Figs 6C and S4B).

To assess whether *Carinh* plays a role in antiviral immunity in vivo, we infected *Carinh*$^{-/-}$ and WT mice with a sublethal dose of influenza A/PR8/34 virus via intranasal inoculation and recorded survival, disease activity score, and weight (Fig 7A). Notably, *Carinh*$^{-/-}$ mice survived significantly longer than their WT counterparts, with more than 83% of *Carinh*$^{-/-}$ mice reaching day 8 post-infection compared with 40% of WT mice (Fig 7B). In addition, *Carinh*$^{-/-}$ mice maintained a higher body weight for longer and presented with a lower disease activity score from days 4 to 7 post-infection (Fig 7C and D), suggesting a reduced inflammatory response that is initially protective. Despite the reduction in disease symptom severity and delayed mortality, all but one *Carinh*$^{-/-}$ mouse succumbed to the infection by day 9, prompting study termination. One WT IAV-infected mouse also showed signs of remission at day 9, whereas mock-treated mice showed no alteration in weight or signs of infection throughout the experiment (Fig 7B–D). Visualization of the distribution of infection foci in the lungs of *Carinh*$^{-/-}$ mice by immunostaining of the IAV

nucleocapsid (Fig 7E, green area) and measurement of whole lung viral *Pr8* RNA content at day 4 post-infection revealed a 2.5-fold higher viral burden in *Carinh*$^{-/-}$ compared with WT mice (Fig 7F). Together, these results suggest that loss of *Carinh* delays the antiviral response to IAV infection in mice, increasing viral burden in the lungs.

To investigate pathways responsible for the delayed immune response to IAV infection in the absence of *Carinh*, we performed RNA-seq on whole lungs at day 4 post-infection. We identified 47 significantly down-regulated and 205 up-regulated genes in *Carinh*$^{-/-}$ mice compared with WT. As expected, *Carinh* was markedly reduced in *Carinh*$^{-/-}$ mice, confirming *Carinh* deficiency in this model (Fig S4C). Ingenuity Pathway Analysis of differentially expressed genes revealed a down-regulation in the IFNβ signaling signature within the positively and negatively differentially expressed genes in the absence of *Carinh* (Fig S4D). Interestingly, in contrast to our results in isolated macrophages (Figs 4F, 5G, and 6C, and S4B), we did not detect significant differences in IRF1 protein in the whole lung of *Carinh*$^{-/-}$ and WT mice by immunostaining or immunoblotting at this time point (Figs 7E and S4E). However, cytometric bead array analysis of antiviral cytokines in nasal shedding samples confirmed that *Carinh*$^{-/-}$ mice had reduced levels of IFNα and IFNβ, as well as the lower expression of the IFN-induced chemokines, MCP1 and CCL5, when compared to nasal samples from WT mice (Fig 7G). Collectively, these data suggest that the absence of *Carinh* in vivo weakens antiviral immunity by interfering with the early IFN response, resulting in higher viral burden.

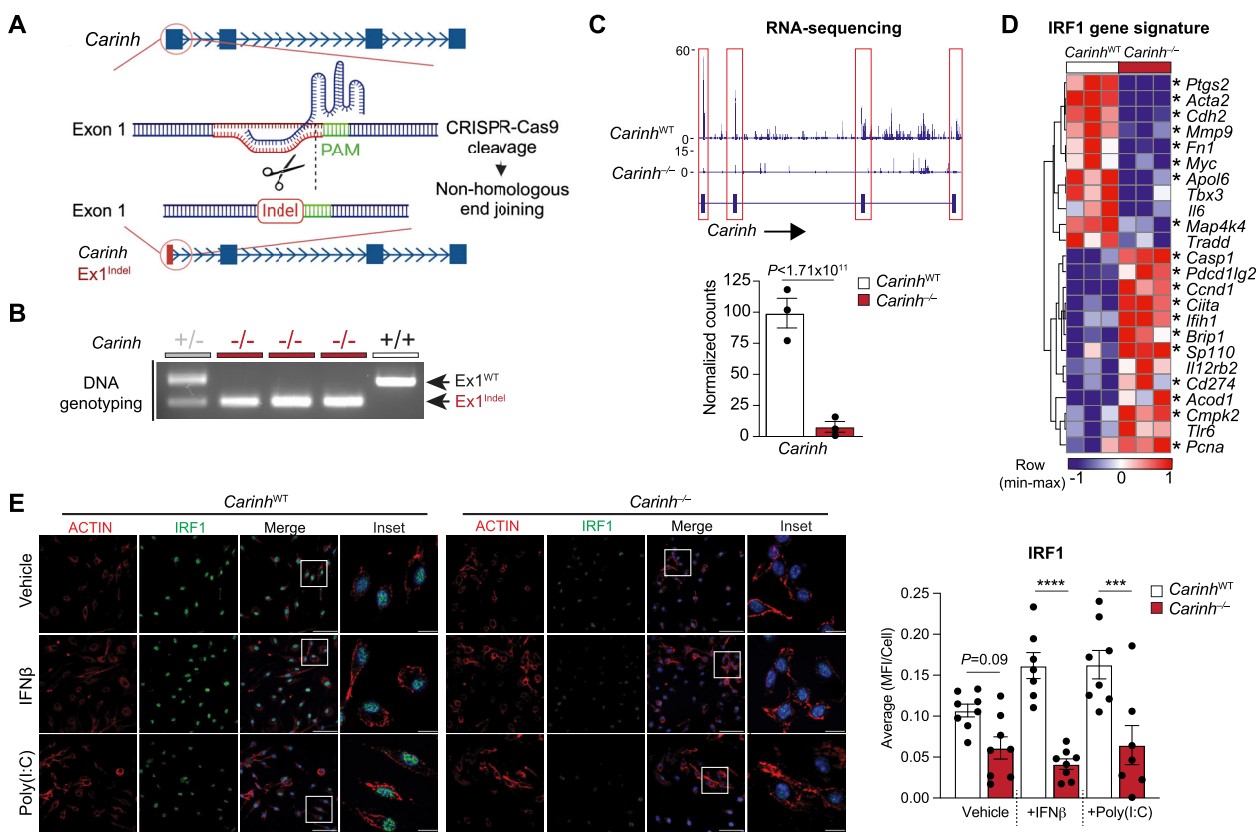

**Figure 6. Generation and validation of *Carinh*^−/− mice.**
**(A)** CRISPR/Cas9 strategy to introduce an indel in the first exon of *Carinh*. **(B)** Genotyping PCR of *Carinh*^WT (+/+), heterozygous (+/−), and *Carinh*^−/− (−/−) mice. **(C)** RNA-seq reads of *Carinh* in BMDM isolated from *Carinh*^WT and *Carinh*^−/− mice. The expression of genes was visualized with Integrative Genomics Viewer 2.9.4. (top) or as normalized counts (bottom). **(D)** Hierarchical clustering heatmap showing Z-scores of genes driven by IRF1 in BMDM isolated from *Carinh*^WT and *Carinh*^−/− mice. **(E)** Immunofluorescent staining of BMDMs isolated from *Carinh*^WT and *Carinh*^−/− mice and treated with a vehicle control, poly(I:C) (1 μg/ml), or IFNβ (1,000 U/ml). Cells were stained for ACTIN (red) and IRF1 (green) with DAPI used for nuclear staining. Scale bars: 50 μm (Merge) and 10 μm (Inset). Quantification of IRF1 is shown at the right; dots are individual fields from independent replicates (n = 3). *P*-values were calculated by DESeq2 (C, D) or one-way ANOVA with prespecified columns (E). *P < 0.05; ***P < 0.001; ****P < 0.0001.

# Discussion

The coordinated expression of ISGs and the type I IFN response is of vital importance to ensure a rapid and efficient immune reaction to respiratory virus infection. It has become apparent that lncRNAs act as key arbitrators of the immune response through pre-, co-, and post-transcriptional regulatory processes; however, mechanistic studies defining roles of lncRNA in the immune system are still limited (Loganathan & Doss, 2023; Mattick et al, 2023). In this study, we establish a role of the human lncRNA *CARINH* in regulating the IFN response and ISG transcription upon viral challenge and demonstrate the conservation of this mechanism in mice. We find that *CARINH* is among the highest up-regulated lncRNAs in the circulation of patients infected with MPV, IAV, or SARS-CoV-2 when compared to healthy controls. Notably, the up-regulation of *CARINH* in human circulation or immune cells infected with a respiratory virus infection coincides with enhanced levels of its proximal gene *IRF1*, which encodes a key transcription factor underlying the interferon response (Panda et al, 2019; Wang et al, 2020; Zhou et al, 2022). In vitro studies recapitulated the coordinated regulation of

the *CARINH*/IRF1 pair by IFNβ and TLR3 signaling, and showed that depletion of *CARINH* reduces IRF1 mRNA and protein in primary macrophages and macrophage cell lines. The impact of *CARINH* is illustrated by the down-regulation of the ISG network when this lncRNA is targeted using antisense oligonucleotides, and a corresponding increase in macrophage viral load upon IAV challenge.

Our work also identifies *Carinh* as the mouse homolog of *CARINH* and shows that its roles in regulating the expression of *IRF1* and downstream ISGs are functionally conserved. Like *CARINH*, *Carinh* is increased upon treatment of BMDMs with the synthetic viral mimic poly(I:C) or directly with IFNAR ligand IFNβ, coincident with up-regulation of the *Irf1* mRNA. Furthermore, knockdown of *Carinh* using antisense oligonucleotides decreases *Irf1* expression and transcriptional activation of an ISRE-reporter gene in macrophages. Using a newly generated *Carinh*^−/− mouse, we show that upon intranasal challenge with IAV, these mice have reduced disease activity markers suggestive of a delayed inflammatory response, which enhances short-term survival compared with WT mice. This is in line with previous studies reporting that a limited inflammatory response upon viral infection in mice may lead to increased virus

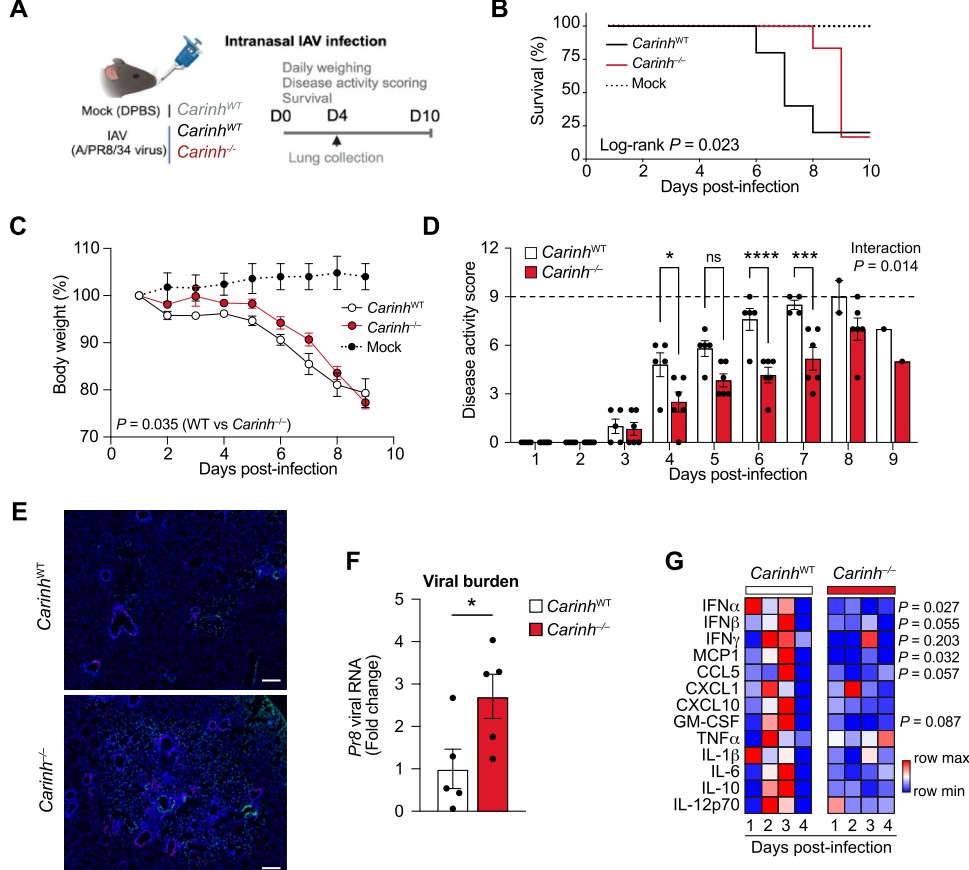

**Figure 7. Absence of *Carinh* in mice weakens antiviral response to IAV infection.**
**(A)** Experimental approach for influenza A/PR8/34 infection via intranasal inoculation in *Carinh*⁻/⁻ or *Carinh*^WT mice. **(B, C, D)** Percentage survival (B), body weights (C), and disease activity score (D) of *Carinh*^WT, *Carinh*⁻/⁻ (n = 5–6/group), and mock-infected mice (n = 2) after challenge with a sublethal dose of influenza A/PR8/34. **(E, F)** Representative microscopic images (E) of IAV nucleocapsid (IAV, green), IRF1 (red), and DNA counterstain (DAPI, blue). **(F)** Scale bar is 200 μm and qRT–PCR analysis of viral *Pr8* RNA (F) in whole lungs and in lungs of *Carinh*^WT and *Carinh*⁻/⁻ 4 d after challenge with a sublethal dose of influenza A/PR8/34. **(G)** Heatmaps of levels of cytokine/chemokine levels in longitudinal nasal shedding samples for 4 d. Data are 4–6 mice/group (B, C, D, E, F, G). *P*-values were calculated by a *t* test (F); by a log-rank test for trend in increased survival from *Carinh*⁻/⁻, to *Carinh*^WT, to mock (B); by repeated-measures two-way ANOVA with significant differences between *Carinh*⁻/⁻ and *Carinh*^WT (C); by repeated-measures two-way ANOVA with Sidak's multiple-comparisons test (D); or by two-way ANOVA with Tukey's multiple-comparisons test between *Carinh*⁻/⁻ and WT mice (G). *P* < 0.05; ***P* < 0.001; *****P* < 0.0001.

titers as opposed to an amplified disease outcome, typically associated with excessive cytokine production (Le Goffic et al, 2006; Szretter et al, 2007; Channappanavar et al, 2016). Despite slower initial weight loss and delayed onset of disease symptoms in *Carinh*⁻/⁻ mice, they eventually succumbed to the viral infection. This suggests an activation of superfluous antiviral immune response pathways in the absence of *Carinh* in vivo. We report that the IFNβ transcriptional signature in whole lungs of *Carinh*⁻/⁻ mice is decreased when compared to WT mice. Kinetic studies have shown that IRF1 induction by IFNβ is required for the initial amplification of the inflammatory response to viral infection, but that sustained antiviral protection relies on type III IFNs (Forero et al, 2019). This is in line with other reports showing that IFNλ is the most up-regulated IFN by low-dose IAV infection (Galani et al, 2017). Interestingly, although IRF1 expression was decreased upon depletion of *CARINH* or *Carinh* in isolated macrophages, total IRF1 expression was unchanged in whole lungs of infected mice at day 4 post-infection, suggesting redundant mechanisms of IRF1 gene regulation at later stages of infection. In support of that, we find that deletion of *Carinh* abrogates the early production of cytokines and chemokines in the upper respiratory tract.

Our studies of *CARINH* corroborate other reports that this lncRNA is an IFN-responsive gene in models of cancer (Huang et al, 2019) and inflammatory bowel disease (Ma et al, 2023; Johnson et al, 2024).

Using ChIRP-seq, we identify *CARINH* binding at multiple gene loci, but it was most enriched at the IRF1 locus, as well as IL18BP, a gene previously shown to be controlled by *CARINH* (Ma et al, 2023). Using transcription reporter assays, we show that *CARINH* contributes to the transcription of ISRE-controlled genes in macrophages, leading to increased ISG expression and amplified IFN secretion after TLR3 agonist or IFNβ treatment. Notably, it has been reported that IRF1 promotes the activation of the NLRP3 inflammasome in macrophages and that cells deficient of IRF1 are more susceptible to infection with IAV in vitro; meanwhile, IRF1⁻/⁻ mice infected with a sublethal dose of IAV do not show noticeable changes in morbidity and body weight (Kuriakose et al, 2018). Findings from another study indicate IRF1⁻/⁻ mice as highly vulnerable to infection with West Nile virus, which in turn pinpoints to effects in macrophages, whereas fibroblasts depleted of IRF1 were not affected by West Nile virus (Brien et al, 2011). It is possible that the immunomodulatory mechanisms driven by IRF1 differ depending on the (viral) stimulus and the cell type and that compensatory mechanisms to overcome potential loss of IRF1 exist, of which *CARINH* could be one.

In esophageal cancer, RNA–protein interaction assays have shown that *CARINH* interacts with ILF3 (Interleukin Enhancer Binding Factor 3) and DHX9 (DExH-Box Helicase 9) to control the expression of the *CARINH* and *IRF1* locus via a feedforward mechanism (Huang et al, 2019). Studies in HeLa cells in the context

of viral infection suggested that the promoter region of *CARINH* has potential enhancer activity, boosting the expression of *IRF1* independently of *CARINH* expression levels (Barriocanal et al, 2022). Interestingly, in a mouse model of inflammatory bowel disease, *Carinh* has been reported to physically interact with p300/CBP, transcriptional co-activators that can bind to a wide set of transcription factors to enhance the expression of numerous target genes, including, but not limited to, *Irf1* (Ma et al, 2023). As lncRNAs can have cell-specific mechanisms of action, further studies of *CARINH* and its role in (cell-specific) IRF1-driven transcriptional programs will be needed to expand on its mechanistic roles in different disease settings.

We demonstrate that both *CARINH* and *IRF1* were highly up-regulated in patients infected with three distinct respiratory viruses and in primary human macrophages infected with IAV. The decreased expression of *IRF1* resulting from targeting *CARINH* with antisense oligonucleotides or genetically in *Carinh*$^{-/-}$ mice resulted in elevated susceptibility to viral infection. Thus, our results further position *CARINH* as an additional layer of IFN pathway regulation through the fine-tuning of *IRF1* expression levels. In support of this important role of *CARINH* in human inflammation, genome-wide association studies and follow-up reporting suggest that the *IRF1* and *CARINH* loci are linked to inflammatory disorders including juvenile idiopathic arthritis (Chiaroni-Clarke et al, 2014), inflammatory bowel disease (Ma et al, 2023; Johnson et al, 2024), and chronic obstructive pulmonary disease (Joo & Himes, 2021; John et al, 2022) where the strength of the immune response to pathogens is impaired (Curtis et al, 2007). Interestingly, IRF1 has been shown to be induced by type II IFNs (e.g., IFNγ), in addition to type I IFNs, to drive corresponding, but distinctly different, immunological activities (Ravi Sundar Jose Geetha et al, 2024). The central function of IFNγ is to augment the immune response upon infection with nonviral pathogens (MacMicking, 2012), and IRF1 deficiency in macrophages causes severe mycobacterial, but not viral, disease in humans (Rosain et al, 2023). Here, we show an impaired expression of IFNγ in human macrophages treated with Gap*CARINH* and in mice depleted for *Carinh* upon viral stimulus, suggesting a potential function for *CARINH* upon bacterial infection. As such, it may be of interest to examine the contribution of *CARINH* in IFNγ-driven macrophage activation upon exposure to bacteria in future studies.

The role of *CARINH*/*Carinh* in regulating the IFN response is in line with a growing body of evidence for functional roles of lncRNAs in the immune response against respiratory viruses. For example, the primate-specific lncRNA *CHROMR* (CHolesterol induced Regulator Of Metabolism RNA) provides another layer of control over the ISG network by sequestering the IRF2-dependent transcriptional co-repressor IRF2BP2 to license transcription of antiviral genes (van Solingen et al, 2022). Similarly, *lncRNA-155*, a lncRNA that stems from the same host gene (*MIR155HG*) as the proinflammatory microRNA miR-155 (O'Connell et al, 2007), stimulates the innate immune response upon IAV infection by inhibiting the expression of PTP1B, a key negative regulator of type I IFN signaling (Maarouf et al, 2019). Other lncRNAs can negatively regulate ISG expression, with *LUCAT1* (Lung Cancer Associated Transcript 1) interacting with STAT1 to restrict JAK-STAT signaling and restore immune homeostasis after the initial inflammatory response to viral infection (Agarwal et al, 2020), and *NRAV* (Negative Regulator of AntiViral response)

reducing the expression of key ISGs, including *MX1*, *IFIT2*, and *IFIT3*, via interaction with transcription factor ZONAB1 and histone modification of target genes (Ouyang et al, 2014). Together with our findings, these reports highlight the emerging roles of lncRNAs in initiating and fine-tuning the type I IFN response and its associated network of ISG that coordinate antiviral innate immunity.

# Materials and Methods

### Mice

All experimental procedures were approved by the New York University School of Medicine's Institutional Animal Care and Use Committee and were conducted in accordance with the US Department of Agriculture Animal Welfare Act and the US Public Health Service Policy on Humane Care and Use of Laboratory Animals. *Carinh*$^{-/-}$ mice were generated by the NYU Rodent Engineering Core. CRISPR/Cas9 technology was used to create an indel in the first exon of *Carinh*, resulting in termination of the *Carinh* transcript. Genotyping was performed to confirm successful insertion of indel. Primers used can be found in Table S4. For influenza virus infection studies, *Carinh* $^{-/-}$ mice or *Carinh* $^{WT}$ littermates were inoculated with a sublethal dose of mouse-adapted influenza A/PR8/34 virus (150 PFU/20 µl/mouse) or sterile DPBS (mock) intranasally under ketamine/xylazine anesthesia at 100 mg/kg/body and 10 mg/kg/body, respectively. Longitudinal nasal shedding samples were collected by dipping the nares of each mouse three times in PBS daily for 4 d. Levels of cytokines and chemokines in nasal shedding samples were quantified using LEGENDplex Mouse Anti-Virus Response Panel (740622; BioLegend) according to the manufacturer's instructions. Body weights and diseases activity score (DAS) were recorded by two blinded observers for 10 d post-infection or until humane endpoint was reached. Briefly, DAS consists of four categories scored 0–3, including lethargy, fur ruffling, hunched posture, and labored breathing (Gonzalez et al, 2018). Humane endpoint was defined as attaining either a 20% body weight loss or a DAS of 9. Lungs were perfused/inflated with PBS and PFA (Thermo Fisher Scientific), followed by 3-d fixation in 4% PFA, incubation with 1 M EDTA (pH 7, Lonza) for 5 d, and then rinsed in serial washes of EtOH before the start of immunohistochemistry as described below. In a second cohort of mice, lungs were collected at day 4 after viral inoculation and homogenized using High-Power Laboratory Homogenizer (Precellys) using CK-14 beads (Precellys) and RNA and protein were isolated as described below.

### Transcriptomic analysis

For differential gene expression analyses between influenza A–infected patients (*n* = 41), human metapneumovirus (*n* = 8), and controls (*n* = 18), we queried publicly available datasets GSE157240 (Tsalik et al, 2021) and GSE190413 (van Solingen et al, 2022) for SARS-CoV-2–infected patients (n = 8) and controls (n = 7) using the R package DESeq2. Differentially expressed mRNA and lncRNA within all three datasets were identified using the lncRNA biotype

annotation within the *Ensembl* gene annotation system (Aken et al, 2016). Pairing of differentially expressed mRNA and lncRNA within 5 kb was achieved through the *GenomicRanges* package in R. Rank-sum ordering of lncRNA was performed based on the highest level of expression of each lncRNA in each disease model compared with their respective control. *CARINH*, *GSEC*, *LINC02422*, *ST3GAL4*, *RESF1*, and *IRF1* expression in human monocyte-derived macrophages infected with influenza A/California/04/09 (H1N1), influenza A/Wyoming/03/03 (H3N2), and influenza A/Vietnam/1203/04 (H5N1), or mock-infected was examined by querying the publicly available dataset GSE97672 (Heinz et al, 2018).

### Cell culture

THP1 cell lines were obtained from the ATCC, and IRF-Lucia luciferase reporter monocytes (THP1-Lucia cells) and RAW-Lucia ISG cells (RAW-Lucia) were obtained from InvivoGen. All cell lines were authenticated using standard ATCC methods (morphology check by microscope, growth curve analysis) and tested monthly for mycoplasma contamination. THP1 cells were maintained in RPMI 1640 (ATCC) supplemented with 10% FBS and 1% penicillin/streptomycin (P/S). THP1-Lucia cells were maintained in RPMI 1640 supplemented 10% FBS, 1% P/S, and 50 µg/ml of Normocin (InvivoGen), and cultured with selectable marker Zeocin (100 µg/ml; InvivoGen) every other passage to maintain stable integration of inducible reporter constructs. THP1 cells and THP1-Lucia cells were differentiated into macrophages in the presence of 100 nM phorbol-12-myristate acetate (PMA, Sigma-Aldrich) for 48–72 h. RAW-Lucia cells were maintained in DMEM (ATCC) supplemented with 10% FBS, 1% P/S, and 50 µg/ml of Normocin, and cultured with selectable marker Zeocin (100 µg/ml) every other passage to maintain stable integration of inducible reporter constructs. Bone marrow–derived macrophages (BMDMs) were prepared by flushing the marrow from the tibiae and femora of 6- to 8-wk-old mice. Cells were differentiated into macrophages in DMEM supplemented with 10% FBS, 1% P/S, and 15% L929-conditioned media for 7 d. Peritoneal macrophages (pMacs) were isolated from mice by peritoneal lavage 3 d after intraperitoneal injection of 1 ml of 3% thioglycolate (Sigma-Aldrich), as previously described (Gallily & Feldman, 1967). Cells were cultured in DMEM with 1% P/S overnight before induction experiments. Human primary PBMCs were isolated from whole blood obtained from the New York Blood Center. Whole blood was processed immediately upon receipt and diluted 1:1 (v:v) with PBS. Ficoll-Paque premium (Sigma-Aldrich) was gently overlaid in SepMate tubes (StemCell) with the diluted blood without breaking the surface plane, followed by centrifugation for 20 min (RT, without brake). The PBMC layer was collected and washed twice in PBS. The cell pellet was diluted in RPMI 1640 supplemented with 10% heat-inactivated FBS, and cell concentration and viability were checked. Monocytes were magnetically labeled with magnetic anti-CD14 microbeads (Miltenyi Biotec) and collected within MACS Column LS (Miltenyi Biotec) in the magnetic field of MACS Separator, according to the manufacturer's instructions. Monocytes were seeded in RPMI 1640 supplemented with 10% FBS and differentiated with 50 ng/ml of recombinant human macrophage–colony-stimulating factor (PeproTech) for 6 d in a humidified incubator at 37°C and 5% $CO_2$. Transient knockdown of *CARINH* was acquired

as follows: PMA-differentiated THP1 cells, PMA-differentiated THP1-Lucia cells, or CD14[+] macrophages were transfected with 50 nM locked nucleic acid GapmeRs (QIAGEN) targeting *CARINH* (Gap-*CARINH*) or Negative Control A (GapCTRL) using Lipofectamine RNAiMax (Life Technologies) as described previously (Hennessy et al, 2019). A similar strategy was used for the transient knockdown of *Carinh* in BMDM and RAW-Lucia cells using GapmeRs targeting *Carinh* (Gap*Carinh*).

### RNA isolation and qRT-PCR

Total RNA was isolated using TRIzol reagent (Invitrogen) and Direct-zol RNA MicroPrep columns (Zymo Research). Upon isolation, RNA was reverse-transcribed using iScript cDNA Synthesis Kit (Bio-Rad Laboratories), and quantitative RT-PCR (qRT-PCR) analysis was conducted using KAPA SYBR Green Supermix (KAPA Biosystems) according to the manufacturer's instructions and quantified on QuantStudio 3 (Applied Biosystems). Fold change in mRNA expression was calculated using the comparative cycle method ($2^{-\Delta\Delta Ct}$) normalized to species-specific housekeeping genes. A list of primers used in this study can be found in Table S4.

### ChIRP sequencing

Cell harvesting, lysis, disruption, and chromatin isolation by RNA purification (ChIRP) were performed as previously described (Chu et al, 2012; van Solingen et al, 2022). A list of probes used in this study can be found in Table S4. DNA and protein were isolated from hybridized magnetic beads followed by DNA sequencing. Briefly, isolated ChIRP DNA was purified via PCR purification columns (Zymo Research) and subjected to Illumina sequencing. Reads were processed using the ChIPseq_PE pipeline from https://github.com/mgildea87/CVRCseq. Read quality was assessed using FastQC (v.0.11.9). Reads were trimmed to remove adapter sequences using fastp (v.0.22.0) (Chen et al, 2018). Trimmed reads were mapped to the *Homo sapiens* reference genome hg38 using bowtie2 (v.2.5.1) (Langmead & Salzberg, 2012). MACS2 (v.2.2.7.1) (Zhang et al, 2008) was run to call peaks in pulldown samples using whole-genome input samples as control. Peaks in ENCODE blacklisted regions were removed based on hg38-blacklist.v2.bed (Amemiya et al, 2019). IDR (v.5.3.1) (Li et al, 2011) was used to identify reproducible peak sets within even and odd *CARINH* pulldown replicates. DiffBind (v.3.8.4) (Stark, 2011) analysis was run on these peak sets to identify regions that were differentially pulled down between anti-*CARINH* samples and anti-LacZ samples (FDR < 0.05 and positive $\log_2$ fold change). ChIRP-seq data are deposited in the Gene Expression Omnibus (accession number GSE275288).

### Cellular response to microbial ligands

To assess the response of macrophages (THP1, CD14[+] monocyte-derived, BMDM, or pMac) to inflammatory cues, we stimulated macrophages or GapmeR-treated macrophages with either 1 µg/ml polyinosinic–polycytidylic acid (poly(I:C), InvivoGen), or 500–1,000 U/ml interferon-beta (IFNβ, human: #IFI014; Millipore; mouse: #124001; Thermo Fisher Scientific) or vehicle control for indicated time periods. After treatment, RNA was isolated and analyzed. Supernatants of GapCTRL- and Gap*CARINH*-treated THP1 stimulated

for 24 h with poly(I:C) and vehicle controls were collected to measure accumulated levels of secreted cytokines. Levels of interferon in supernatants were quantified using LEGENDplex Human Type 1/2/3 Interferon Panel (740396; BioLegend) according to the manufacturer's instructions. Cytotoxicity was determined with LDH-Glo Cytotoxicity Assay (Promega) according to the manufacturer's instructions and by cellular staining using propidium iodide (P3566; Life Technologies) and annexin V (A13199; Invitrogen).

### RNA FISH

Custom Stellaris FISH Probes were designed against *CARINH* using Stellaris FISH Probe Designer (LGC Biosearch Technologies). Formaldehyde-fixed THP1 macrophages were permeabilized with 70% isopropanol and subsequently simultaneously hybridized with the *CARINH* Stellaris FISH Probe set labeled with Quasar 670 Dye (LGC Biosearch Technologies).

### Gene expression profiling

RNA was isolated from THP1 macrophages transfected with 50 nM locked nucleic acid GapmeRs (QIAGEN) targeting *CARINH* (Gap-*CARINH*) or Negative Control A (GapCTRL). RNA was subsequently reverse-transcribed, and qRT-PCR analysis of type I interferon response genes was performed using RT² Profiler PCR Arrays (PAHS-016ZA; QIAGEN) according to the manufacturer's protocol. Data analysis was performed using the manufacturer's integrated Web-based software package of the PCR Array System using ΔΔCt-based fold change calculations.

### Western blot analysis

Proteins were isolated in radioimmunoprecipitation buffer (RIPA) from isolated lungs, THP1 macrophages, or BMDM transfected with 50 nM locked nucleic acid GapmeRs (QIAGEN) targeting *CARINH*, *Carinh*, or a GapmeR control. Protein concentration was determined by BCA measurement (Pierce), and 5 $\mu$g total protein per sample was loaded in a Wes automated Western blot system (Bio-Techne) and assayed for the expression of IRF1 (ab186384; Abcam) and housekeeping protein GAPDH (#2118S; Cell Signaling Technologies). In this automated system, samples are denatured in a proprietary dithiothreitol solution, immobilized, immunoassayed, and imaged in individual capillaries in the instrument.

### Quantification of influenza A virus infection in THP1 macrophages

THP1 macrophages transiently knocked down for *CARINH* were infected with 70,000 plaque-forming units (PFU, as determined on MDCK cells) of influenza A/WSN/1933 (H1N1) virus per 70,000 cells (MOI of 1). The virus inoculate was diluted in DPBS supplemented with calcium and magnesium. Cell growth media were replaced by virus dilution and incubated for 1 h at 37°C and 5% $CO_2$. After 1 h, the virus was aspirated, RPMI 1640 with 20% FBS was added to the cells, and cells were incubated at 37°C and 5% $CO_2$. At 24 h, the cells were fixed with 8% PFA (Thermo Fisher Scientific), quenched with 50 mM $NH_4Cl$, and washed with PBS. Cells were stained with a monoclonal mouse anti-NP antibody (MAB8251; Sigma-Aldrich) followed by anti-

mouse Alexa 488 secondary antibody (R37120; Thermo Fisher Scientific) and nuclear staining DAPI (Sigma-Aldrich). Cells were washed with PBS leaving the last wash on before imaging. Plates were imaged using the Cell-Insight CX7 high-content screening platform. Images were analyzed and quantified with HCS Navigator software for total and infected cell numbers.

### Luciferase reporter assay

PMA-treated THP1-Lucia cells or RAW-Lucia ISG cells were transfected with GapmeRs targeting *CARINH* or *Carinh*, respectively, or a GapmeR control as described above; 24 h post-transfection, the THP1-Lucia/RAW-Lucia ISG cells were treated with poly(I:C), IFN$\beta$, or a vehicle control, as described above. Supernatants were taken on indicated time points, and activation of the interferon regulatory factor (IRF) at the ISRE was measured by detecting luciferase levels in the supernatants using QUANTI-Luc (InvivoGen). Detected levels of luciferase at the start of the experiment (0 h) were set to 100%.

### Immunohistochemistry

BMDMs and pMacs were plated on #1.5 thick round coverslips (Thermo Fisher Scientific) before treatment. After 8 h of treatment with poly(I:C), IFN$\beta$, or vehicle control, cells were fixed in 4% PFA, blocked and permeabilized in solution containing 5% normal goat serum and 0.2% Triton X-100, stained overnight at 4°C with an antibody against IRF1 (ab230652; Abcam), and followed by staining with a fluorescent secondary antibody (Alexa Fluor 488, A-11008; Invitrogen) and a fluorescent-conjugated F-actin probe (Alexa Fluor 555 Phalloidin, A34055; Thermo Fisher Scientific) for 1 h at RT. Washes between steps were done accordingly, and DAPI was used for nuclear staining. Cells were mounted and visualized using a Zeiss 700 confocal microscope and imaged at 63X (numerical aperture 1.4, oil lens). IRF1 protein expression (MFI/Cell) was quantified in multiple fields of view from independent wells using CellProfiler (version 4.2.8), incorporating the object processing module to segment cellular compartments (total versus nuclear) and the measurement module to measure fluorescent intensity. Paraffin-embedded lung sections were immunostained on Leica Bond RX, according to the manufacturer's instructions. In brief, deparaffinized sections underwent a 20-min heat retrieval in Leica ER2 buffer (pH 9, AR9640) followed by Rodent Block (RBM961 L; Biocare) before a 1-h incubation with IAV NP protein antibody (Mouse Anti-Influenza A, Nucleoprotein-UNLB, Cat #10780-01; Southern Biotech) and an antibody against mouse IRF1 (Cat #8478, 1:50; Cell Signaling). Slides were counterstained with DAPI. Semi-automated image acquisition was performed on a Vectra Polaris multispectral imaging system. After whole slide scanning at 20X, the tissue was manually outlined to select fields for spectral unmixing and image analysis using InForm version 2.6 software from Akoya Biosciences. Research image data were managed using OMERO Plus v5.6 (Glencoe Software).

### RNA sequencing

Upon isolation, RNA was used to generate barcoded cDNA libraries using the TruSeq RNA Sample Preparation kit (Illumina). Indexed libraries were pooled and sequenced (paired-end 50-bp reads) on

the Illumina HiSEQ 2500 platform. Quality control of sequencing reads was assessed using FastQC (v.0.11.7). Reads were mapped to mouse reference genome mm39 using STAR (v2.6.1d), and genomic features were then assigned using Subread featureCounts (v.1.6.3). Raw counts were normalized, and differential expression analysis was performed in R using DESeq2 (v.1.30.1). RNA-seq data are deposited in the GEO under accession numbers GSE247501 and GSE261123. The expression of genes was visualized using Integrative Genomics Viewer 2.9.4. Downstream analysis was performed using Ingenuity Pathway Analysis (QIAGEN).

### Statistics

Statistical significance between two groups of independent biological replicates was evaluated with a *t* test or a one-sample *t* and Wilcoxon test. One-way ANOVA was performed when comparing three groups or more for univariate comparisons. Repeated-measures two-way ANOVA was used when comparing two groups or more for bivariate analyses, and F-statistics were performed to examine interactions. Dunnett's post hoc multiple-comparisons test (MCT) was used when comparing to a control group, Tukey's post hoc MCT was used to compare all groups if either the ANOVA group or group × time interaction was significant, and Sidak's post hoc MCT was used when comparing a series of groups selected a priori. Statistical and correlation analyses were performed using GraphPad Prism software. The threshold for statistical significance was $P \leq 0.05$. All quantitative data are presented as the mean ± SEM.

## Data Availability

RNA-seq and ChIRP-seq datasets have been deposited in the Gene Expression Omnibus and are available under accession numbers GSE247501, GSE261123, and GSE275288. All other data are included in the article and/or Supplemental Material.

## Supplementary Information

## Acknowledgements

*Carinh*$^{-/-}$ mice were generated by NYU Rodent Engineering Core. This work was supported by grants from the Canadian Institutes of Health Research (MFE-176524 to Y Cyr), the National Institutes of Health (R35HL135799 and P01HL131481 to KJ Moore, and R01AI143639 and R21AI139374 to M Dittmann), and the American Heart Association (19CDA34630066 and 23SCEFIA1153739 to C van Solingen, 23POST1029885 to M Gourvest, and 915560 to AAC Newman).

### Author Contributions

Y Cyr: conceptualization, data curation, formal analysis, investigation, visualization, methodology, project administration, and writing—original draft, review, and editing.

M Gourvest: data curation and formal analysis.
GO Ciabattoni: data curation and formal analysis.
T Zhang: data curation and formal analysis.
AAC Newman: data curation and formal analysis.
T Zahr: data curation and formal analysis.
S Delbare: data curation and formal analysis.
F Schlamp: data curation and formal analysis.
M Dittmann: data curation, formal analysis, and funding acquisition.
KJ Moore: data curation, formal analysis, supervision, funding acquisition, and writing—original draft, review, and editing.
C van Solingen: conceptualization, data curation, formal analysis, supervision, funding acquisition, investigation, visualization, methodology, project administration, and writing—original draft, review, and editing.

### Conflict of Interest Statement

KJ Moore is on the Scientific Advisory Board of Beren Therapeutics and Bitterroot Bio. The other authors declare no conflict of interest.

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
