## [Reviewer comments · Life Science Alliance]

Life Science Alliance

LncRNA CARINH regulates expression and function of innate transcription factor IRF1 in macrophages

Yannick Cyr, Morgane Gourvest, Grace Ciabattoni, Tracy Zhang, Alexandra Newman, Tarik Zahr, Sofie Delbare, Florencia Schlamp, Meike Dittmann, Kathryn Moore, and Coen van Solingen

DOI: <https://doi.org/10.26508/lsa.202403021>

Corresponding author(s): Coen van Solingen, New York University

Review Timeline:

Submission Date:	2024-08-28
Editorial Decision:	2024-10-15
Revision Received:	2024-12-03
Editorial Decision:	2024-12-24
Revision Received:	2024-12-27
Accepted:	2024-12-30

Transaction Report:

October 15, 2024

Re: Life Science Alliance manuscript #LSA-2024-03021-T

Dr. Coen van Solingen
New York University Grossman School of Medicine
NYU Cardiovascular Research Center
435 E 30th Street
Science Building 704
New York, NY 10026

Dear Dr. van Solingen,

Thank you for submitting your manuscript entitled "LncRNA IRF1-AS1 regulates expression and function of innate transcription factor IRF1 in macrophages" to Life Science Alliance. The manuscript was assessed by expert reviewers, whose comments are appended to this letter. We invite you to submit a revised manuscript addressing the Reviewer comments.

Thank you for this interesting contribution to Life Science Alliance. We are looking forward to receiving your revised manuscript.

Sincerely,

B. MANUSCRIPT ORGANIZATION AND FORMATTING:

Reviewer #1 (Comments to the Authors (Required)):

This is a very interesting and comprehensive study by Cyr and colleagues characterizing the regulation and function of lncRNA IRF1-AS1 and its murine ortholog Gm12216 in regulating the IFN transcriptional program upon viral infection. The authors identified IRF1-AS1 and its proximal coding gene IRF1 as a putative cis-acting lncRNA-mRNA pair induced by multiple viral infections in human cells especially CD14+ monocytes. Further studies identified this lncRNA-mRNA pair as being IFN and polyIC responsive as well. Elegant Hi-C analysis revealed specific contacts between IRF1-AS1 and the IRF1 gene locus. This was complemented by ChIRP analysis which also defined interactions between IRF1-AS1 and intronic regions in IRF1 and IL18BP a gene previously known to be regulated by IRF1-AS1. Functional studies with GapmeR antisense oligonucleotides targeting IRF1-AS1 revealed that the expression of IRF1 was reduced in CD14+ monocytes. As a consequence, since IRF1 contributes to the ISG response, the authors profiled the transcript levels of a selection of ISGs in THP1 macrophages using a qRT-PCR array. Compared to GapCTRL, GapIRF1-AS1 treatment significantly reduced the expression of more than 25% of ISGs examined including OAS1, OAS2, IL6, and IFIT2, IFIT3.

Studies in macrophages with influenza A/WSN/1933 (H1N1) virus also revealed IRF1-AS1 as a critical regulator of IRF1 expression and ISGs required to limit viral infection in human innate immune cells.

Furthermore, the authors leveraged Crispr systems to characterize Gm12216 a mouse ortholog of IRF1-AS1 defining conserved functions in regulating macrophage innate immune responses through control of *Irf1* expression. Studies of mice lacking Gm12216 subjected to IAV infection revealed in vivo weakened antiviral immunity by interfering with the early IFN response, resulting in higher viral burdens. However, a concomitant milder inflammatory response enabled mice lacking Gm12216 in vivo to survive longer than their WT counterparts.

Overall, I found this to be a very comprehensive, elegant and carefully performed series of studies. The studies are well controlled and the data is rigorous and well supported. The comparative analysis of mice and humans and the elucidation of conserved functions of a lncRNA such as IRF1-AS1 is noteworthy. This study adds to our understanding of the emerging roles of long noncoding RNAs in initiating and fine tuning the type I IFN response and its associated network of ISG that coordinate antiviral innate immunity.

Reviewer #2 (Comments to the Authors (Required)):

Long noncoding RNAs (lncRNAs) are understudied regulators of gene expression. The authors investigate the role of one human lncRNA, IRF1-AS1 and its murine orthologue, Gm12216, in immunity against influenza A virus (IAV). Both lncRNAs overlap the *Irf1* locus in antisense direction. The data suggest that knockdown of the lncRNA reduces expression of IRF1, a regulator of interferon-stimulated genes (ISG), and the ability of cells to limit IVA growth. Deletion of Gm12216 in the mouse genome results in increased viral load, but also in increased survival and reduced disease scores of the infected animals.

General comments:

1. The manuscript presents IRF1 as the main target of the lncRNA and this interpretation can be sustained as long as the study is limited to cells. The in vivo experiments strongly argue for other important targets as their most likely explanation is that Gm12216 promotes virus-induced inflammation. This notion is supported by effects of the lncRNA on many other genes including IL-18BP and the GM12216-dependent proinflammatory genes in fig. 7G. What is generally missing, but absolutely essential for the conclusions, is a comparison of the Gm12216 or IRF1-AS1 LOF experiments with IRF1 LOF.
2. The authors present lncRNA-regulated IRF1 expression as important for the expression of type I and type III IFN. This is inconsistent with a wealth of literature opposing the original notion of IRF1 as a positive regulator of IFN synthesis. This role has been attributed to IRF3 and IRF7. Again, an IRF1 LOF experiment is needed to show (or refute) its role as a regulator of IFN synthesis.
3. The paper places IRF1 in the response to type I IFN. Many studies including this recent paper (<https://doi.org/10.1038/s44318-024-00092-7>) demonstrate that IRF1 plays a more important role in the response to IFN γ . Consistent with this, mutation of the *Irf1* gene in humans causes mendelian susceptibility to mycobacterial disease, a typical

feature resulting from the lack of IFN γ activity (DOI: 10.1016/j.cell.2022.12.038). Figure 7G shows that IFN γ synthesis is reduced in IAV-infected Gm12216 $^{-/-}$ mice. The paper would benefit from investigation of IFN γ -dependent immunity in infection models such as *L.monocytogenes*.

Specific comments:

1. On what basis does fig. 6D show an IRF1 gene signature? If anything, it is very incomplete as prominent IRf1 targets are missing.
2. The knockdown of Gm12216 in BMDM (fig. 5F) is very incomplete and it is questionable whether the corresponding reduction of Irf1 mRNA expression is biologically meaningful. The legend doesn't explain whether steady-state or induced mRNA has been analyzed. Similarly, the knockdown of IRF1-AS1 produces a minor effect on IRF1 expression (fig. 4F).

Reviewer #3 (Comments to the Authors (Required)):

Comment to the authors

The manuscript titled " LncRNA IRF1-AS1 regulates expression and function of innate transcription factor IRF1 in human and mouse macrophages" by Cyr et al. demonstrated the impact of the IRF1-AS1 lncRNA gene, by mediating IRF1 gene, on viral infection. The authors focused on the novel lncRNA IRF1-AS1's effect on viral infection, which was identified from omics datasets of patients. Despite the gene name containing IRF1, its role has been elusive. The authors analyzed the functionality of this gene and its mouse homolog, the Gm12216 gene, by creating knockout mice and conducting in vivo infection models, which demonstrated its role in IRF1 gene expression. Despite its novelty of data, the manuscript seems to lack substantial explanation, especially regarding the motivation for each experiment, making it somewhat confusing. Additionally, one critical piece of data for the logical flow is missing. The authors should include more explanations in the text to improve overall clarity.

-Additional experiment required-

Fig.6

Please provide the evidence that shows a decrease of IRF1 expression in cells from Gm12216 $^{-/-}$ mice, as the evidence provided so far is based on in vitro knockdown, which may not accurately represent in vivo cells. The authors should thoroughly search for such cells and analyze them as shown in Fig. 6D. For instance, peripheral monocytes or alveolar macrophages, which are known to play a crucial role in IAV infection, would be suitable candidates. Considering that Gm12216 could be inducible upon interferon signaling, interferon- or poly(I: C)-treated BMDM might be effective.

-Additional issues to be addressed-

Fig.4

The term "Antisense" may create a negative impression for readers regarding the expression of the target gene, leading to confusion. The authors need to provide a thorough explanation with citations as to why knocking down this lncRNA is necessary to understand its functionality, especially considering that natural antisense RNA may play a role in stabilizing the target RNA, prior to knockdown experiments using GampeR technology. If the authors agree, it would be beneficial to use an alternative gene name, such as CARINH, consistently throughout the text to avoid such confusion.

Fig.7

The survival curve in Fig. 7B and the viral load in Fig. 7F appear to be somewhat conflicting. This may indicate that Gm12216 $^{-/-}$ cells have a reduced ability to produce cytokines, resulting in mice not experiencing a cytokine storm upon infection. Instead, this led to a delay in virus clearance, as demonstrated in Fig. 7F. The authors should provide some explanations or speculations for this, including references to previous studies that have reported similar results.

Fig.2a

Pearson collinearity analysis is unsuitable because raw CPM or FPKM values do not follow a normal distribution. Taking the logarithm of the CPM/FPKM values before plotting is recommended.

Fig.2b,2c,4g,5h,7c,7d

When conducting ANOVA, it is mandatory to provide F-statistics. Two-way ANOVA should be used to examine the interaction of two factors, and the author should use the correct statistical methods or provide statements regarding such interactions.

Fig.3b, 4b, 5d, 5e, 5f, 5g

Why were the values in the Vehicle group the same? It's unlikely that all values are identical. A Student t-test is inappropriate here because the Vehicle group variance is 0.

Fig.5c is unnecessary because the cells used in this dataset are irrelevant to this manuscript.

Fig.6c, 6d

The statistical significance in RNA-seq data should be determined using negative binomial-based statistics, such as DESeq2, rather than the Student t-test.

Reviewer #1:**Comments:**

This is a very interesting and comprehensive study by Cyr and colleagues characterizing the regulation and function of lncRNA IRF1-AS1 and its murine ortholog Gm12216 in regulating the IFN transcriptional program upon viral infection. The authors identified IRF1-AS1 and its proximal coding gene IRF1 as a putative cis-acting lncRNA-mRNA pair induced by multiple viral infections in human cells especially CD14+ monocytes. Further studies identified this lncRNA-mRNA pair as being IFN and polyIC responsive as well. Elegant Hi-C analysis revealed specific contacts between IRF1-AS1 and the IRF1 gene locus. This was complemented by ChIRP analysis which also defined interactions between IRF1-AS1 and intronic regions in IRF1 and IL18BP a gene previously known to be regulated by IRF1-AS1. Functional studies with GapmeR antisense oligonucleotides targeting IRF1-AS1 revealed that the expression of IRF1 was reduced in CD14+ monocytes. As a consequence, since IRF1 contributes to the ISG response, the authors profiled the transcript levels of a selection of ISGs in THP1 macrophages using a qRT-PCR array. Compared to GapCTRL, GapIRF1-AS1 treatment significantly reduced the expression of more than 25% of ISGs examined including OAS1, OAS2, IL6, and IFIT2, IFIT3. Studies in macrophages with influenza A/WSN/1933 (H1N1) virus also revealed IRF1-AS1 as a critical regulator of IRF1 expression and ISGs required to limit viral infection in human innate immune cells.

Furthermore, the authors leveraged Crispr systems to characterize Gm12216 a mouse ortholog of IRF1-AS1 defining conserved functions in regulating macrophage innate immune responses through control of Irf1 expression. Studies of mice lacking Gm12214 subjected to IAV infection revealed in vivo weakened antiviral immunity by interfering with the early IFN response, resulting in higher viral burdens. However, a concomitant milder inflammatory response enabled mice lacking Gm12216 in vivo to survive longer than their WT counterparts.

Overall, I found this to be a very comprehensive, elegant and carefully performed series of studies. The studies are well controlled and the data is rigorous and well supported. The comparative analysis of mice and humans and the elucidation of conserved functions of a lncRNA such as IRF1-AS1 is noteworthy. This study adds to our understanding of the emerging roles of long noncoding RNAs in initiating and fine tuning the type I IFN response and its associated network of ISG that coordinate antiviral innate immunity.

> We thank the reviewers for their positive comments and feedback on our manuscript.

Reviewer #2:**Comments:**

Long noncoding RNAs (lncRNAs) are understudied regulators of gene expression. The authors investigate the role of one human lncRNA, IRF1-AS1 and its murine orthologue, Gm12216, in immunity against influenza A virus (IAV). Both lncRNAs overlap the Irf1 locus in antisense direction. The data suggest that knockdown of the lncRNA reduces expression of IRF1, a regulator of interferon-stimulated genes (ISG), and the ability of cells to limit IAV growth. Deletion of Gm12216 in the mouse genome results in increased viral load, but also in increased survival and reduced disease scores of the infected animals.

General comments:

1. The manuscript presents IRF1 as the main target of the lncRNA and this interpretation can be sustained as long as the study is limited to cells. The in vivo experiments strongly argue for other important targets as their most likely explanation is that Gm12216 promotes virus-induced

inflammation. This notion is supported by effects of the lncRNA on many other genes including IL-18BP and the GM12216-dependent pro-inflammatory genes in fig. 7G. What is generally missing, but absolutely essential for the conclusions, is a comparison of the Gm12216 or IRF1-AS1 LOF experiments with IRF1 LOF.

> We would like to thank the reviewer for this thoughtful comment. Although, we have found a clear correlation between IRF1-AS1 (henceforth: CARINH, per comments to Reviewer 3) and IRF1 in humans in three different types of respiratory tract infections, we agree that our experiments in mice do not perfectly mirror the mechanistic findings *in vitro*. We now address this in the revised discussion. [Discussion, Page 11/12: Kinetic studies... respiratory tract]. We discuss how knock-out (KO) of Gm12216/Carinh *in vivo* does not phenocopy *Irf1*^{-/-} in mice upon viral challenge [Discussion, Page 12: Notably, it has been... West Nile Virus], which is perhaps not surprising since the Gm12216/Carinh KO does not result in complete ablation of IRF expression. These data suggest that compensatory mechanisms to overcome IRF1 attenuation exist to protect the innate immune response to viruses.

As requested, we performed IRF1-loss-of-function studies using siRNA to deplete IRF1 in THP-1 macrophages and assayed mRNA expression of 84 ISG by RT2-profiler as shown in Fig 4C for CARINH. Despite a significant decrease of IRF1 upon transfection of siIRF1 (Log2FC=-1.23, P< 7.1308E-05, n=3), we did not observe marked changes in ISG gene expression (Reviewer Fig. 1A). We also treated THP1-Dual reporter cells with siIRF1 (or siCTRL) and stimulated with poly(I:C) or IFNβ (Reviewer Fig. 1B). Again, we observed that there are compensatory mechanisms that allow for transcriptional activation of ISRE-containing promoters in IRF1-depleted cells.

Reviewer Fig. 1. A qPCR array-based gene expression profiling of 84 type I interferon response genes in THP1 macrophages treated with siIRF1 versus siCTRL. B. Reporter assay for ISRE-driven transcription in human THP1-Lucia reporter macrophages transfected with siIRF1 or siCTRL and treated with or poly(I:C) (1 µg/mL) or IFNβ (500 U/mL). Relative luciferase expression (relative units [RU]) is normalized to time 0h, set at 100%.

Overall, our data suggest that CARINH acts as a rheostat that tunes IRF1 expression and function, consistent with the role of many long noncoding RNAs. We have clarified our discussion on page 12 and 13 (highlighted) as follows: 'As lncRNAs can have cell specific mechanisms of action, further studies of CARINH in the context of IRF1-biology and different disease settings will be needed to further clarify its mechanistic roles' and 'Thus, our results further position CARINH as an additional layer of IFN pathway regulation through fine-tuning of IRF1 expression levels'.

2. The authors present lncRNA-regulated IRF1 expression as important for the expression of type I and type III IFN. This is inconsistent with a wealth of literature opposing the original notion of IRF1 as a positive regulator of IFN synthesis. This role has been attributed to IRF3 and IRF7. Again, an IRF1 LOF experiment is needed to show (or refute) its role as a regulator of IFN synthesis.

> We would like to thank the reviewer for pointing out this inconsistency. We have revised our manuscript to indicate that although *CARINH* and *IRF1* are activated and induced by interferons, the positive feedback loop to amplify the synthesis of IFN goes through the upregulation of ISGs, which indirectly supports the production of *IRF3/7* to further promote the expression of IFNs and enhance the immune response.

3. The paper places *IRF1* in the response to type I IFN. Many studies including this recent paper (<https://doi.org/10.1038/s44318-024-00092-7>) demonstrate that *IRF1* plays a more important role in the response to IFN γ . Consistent with this, mutation of the *Irf1* gene in humans causes mendelian susceptibility to mycobacterial disease, a typical feature resulting from the lack of IFN γ activity (DOI: 10.1016/j.cell.2022.12.038). Figure 7G shows that IFN γ synthesis is reduced in IAV-infected *Gm12216*^{-/-} mice. The paper would benefit from investigation of IFN γ -dependent immunity in infection models such as *L. monocytogenes*.

> We thank the reviewer for pointing out these recent interesting studies. We have now revised our discussion to describe *IRF1*'s role in the response to IFN γ (page 13 – highlighted) While we agree that future experiments using bacterial infection models like *L. monocytogenes* are of interest, the scope of our current manuscript focuses on the role of this lncRNA in viral infections.

Specific comments:

1. On what basis does fig. 6D show an *IRF1* gene signature? If anything, it is very incomplete as prominent *IRF1* targets are missing.

> Thank you for the comment – it is an important point to clarify, which we have done in the text related to **Fig. 6D**. The *IRF1* gene signature used was derived from the unbiased Ingenuity Knowledge Base from Qiagen's Ingenuity Pathway Analysis software. This software uses genes (based on FC and p value) in a dataset to extrapolate common 'Upstream Regulators'. This Upstream Regulator Analysis is based on expected causal effects between upstream regulators and targets, derived from the literature compiled in the Ingenuity Knowledge Base. This analysis examines the known targets of each upstream regulator in a dataset, compares the targets actual direction of change to expectations derived from the literature, then issues a prediction for each upstream regulator. The direction of change is the gene expression in the experimental samples relative to a control (Krämer et al, Bioinformatics 2013).

To further support this unbiased and causal network driven by *IRF1*, we used StringDB (as described here: Szklarczyk et al, NAR 2023) to cluster the genes in the *IRF* gene signature, and we noted that all but two genes cluster strongly together (**Reviewer Fig. 2**). We have provided further explanation of the basis of the *IRF1* gene signature in the results section where we discuss **Fig. 6D** (Page 9, highlighted).

Reviewer Fig. 2. StringDB analysis showing clustering of genes present in IRF1 gene signature derived from Ingenuity Knowledge Base (Qiagen).

2. The knockdown of *Gm12216* in BMDM (fig. 5F) is very incomplete and it is questionable whether the corresponding reduction of *Irf1* mRNA expression is biologically meaningful. The legend doesn't explain whether steady-state or induced mRNA has been analyzed. Similarly, the knockdown of *IRF1-AS1* produces a minor effect on *IRF1* expression (fig. 4F).

> The reviewer notes correctly that the knockdown of *Carinh* and *CARINH* is reduced only by ~25% (Fig. 5E/F) and ~50% (Fig. 4B/F), respectively, from starting levels. Transfection of GapmeRs into macrophages is notoriously challenging and our levels of knockdown are consistent with other studies. However, as shown in Fig. 5G and Fig. 4G, even this incomplete knockdown of *Carinh* and *CARINH* leads to a biologically meaningful decrease in ISRE activity upon treatment with polyIC, or IFN β , but not with vehicle (Fig. EV3A, D). Moreover, human macrophages treated with GapmeRs and subsequently infected with influenza demonstrate an increase in infection upon depletion of *CARINH* as clear indication of biological relevance (Fig. 4H) of knockdown of *CARINH*.

Reviewer #3:

Comments:

The manuscript titled "LncRNA *IRF1-AS1* regulates expression and function of innate transcription factor *IRF1* in human and mouse macrophages" by Cyr et al. demonstrated the impact of the *IRF1-AS1* lncRNA gene, by mediating *IRF1* gene, on viral infection. The authors focused on the novel lncRNA *IRF1-AS1*'s effect on viral infection, which was identified from omics datasets of patients. Despite the gene name containing *IRF1*, its role has been elusive. The authors analyzed the functionality of this gene and its mouse homolog, the *Gm12216* gene, by creating knockout mice and conducting in vivo infection models, which demonstrated its role in *IRF1* gene expression. Despite its novelty of data, the manuscript seems to lack substantial explanation, especially regarding the motivation for each experiment, making it somewhat confusing. Additionally, one critical piece of data for the logical flow is missing. The authors should include more explanations in the text to improve overall clarity.

> We thank the reviewer for their comments on our manuscript. We have revised the text to further clarify the motivation for each experiment as requested.

-Additional experiment required- Fig.6

Please provide the evidence that shows a decrease of *IRF1* expression in cells from *Gm12216*^{-/-} mice, as the evidence provided so far is based on in vitro knockdown, which may not accurately represent in vivo cells. The authors should thoroughly search for such cells and analyze them as shown in Fig. 6D. For instance, peripheral monocytes or alveolar macrophages, which are known to play a crucial role in IAV infection, would be suitable candidates. Considering that *Gm12216* could be inducible upon interferon signaling, interferon- or poly(I:C)-treated BMDM might be effective.

> As requested, we include new data showing a decrease in *IRF1* expression in cells from *Carinh*^{-/-} (*Gm12216*^{-/-}) mice. As shown below, immunohistochemical staining for *IRF1* in BMDMs and peritoneal macrophages (pmacs) treated with poly(I:C), IFN β or vehicle, show reduced levels of *IRF1* in *Carinh*^{-/-} compared to WT mice (new Fig. 6E and new Fig. EV4B).

New Fig. 6E, New Fig. EV4B Immunofluorescent staining of BMDMs (6E) and peritoneal macrophages (EV4B) isolated from *Carinh^{WT}* and *Carinh^{-/-}* mice and treatment with vehicle control, poly(I:C) (1 μ g/mL) or IFN β (1000 U/mL) for ACTIN (red) and IRF1 (green) with DAPI-stained nuclei. Scalebars: 50 μ m (Merge) and 10 μ m (Inset). Quantification of IRF1 is shown at the right; dots are individual fields of view from three individual experiments (n=3).

<Major issues>

Fig.4

The term "Antisense" may create a negative impression for readers regarding the expression of the target gene, leading to confusion. The authors need to provide a thorough explanation with citations as to why knocking down this lncRNA is necessary to understand its functionality, especially considering that natural antisense RNA may play a role in stabilizing the target RNA, prior to knockdown experiments using GampeR technology. If the authors agree, it would be beneficial to use an alternative gene name, such as CARINH, consistently throughout the text to avoid such confusion.

> We thank the reviewer for pointing out the potential negative impression for readers regarding the term 'antisense' and have changed the annotation of the gene names of *IRF1-AS1* and *Gm12216* to *CARINH* and *Carinh*, respectively as recently described by Ma *et al* in 2023. This is in line with the current nomenclature as reported by HUGO (https://www.genenames.org/data/gene-symbol-report/#!/hgnc_id/HGNC:33838).

Fig.7

The survival curve in Fig. 7B and the viral load in Fig. 7F appear to be somewhat conflicting. This may indicate that *Gm12216*^{-/-} cells have a reduced ability to produce cytokines, resulting in mice not experiencing a cytokine storm upon infection. Instead, this led to a delay in virus clearance, as demonstrated in Fig. 7F. The authors should provide some explanations or speculations for this, including references to previous studies that have reported similar results.

> As suggested by the reviewer, we have clarified this important point in the discussion. We now discuss how the decreased cytokine levels in the *Carinh^{-/-}* mice upon influenza infection initially protects them from the onset of early disease symptoms leading to short term survival when compared to WT littermates. However, ultimately *Carinh^{-/-}* mice do succumb to the virus infection. This is now described on page 11 of the discussion and pasted below, as we include references to studies that reported similar results.

'This is in line with previous studies reporting that a limited inflammatory response upon viral infection in mice lead to increased virus titers and not to severe disease outcome that is typically associated with excessive cytokine production (Channappanavar et al, 2016; Le Goffic et al, 2006; Szretter et al, 2007).'

<Minor issues>

Fig.2a

Pearson collinearity analysis is unsuitable because raw CPM or FPKM values do not follow a normal distribution. Taking the logarithm of the CPM/FPKM values before plotting is recommended.

> We would like to thank the reviewer for their helpful comments and have changed **Fig. 2A** to show the logarithm of the CPM/FPKM values before replottting the data presented. Rho and P values were recalculated and updated in **Fig. 2A** We have also recalculated the Rho and P values for the mRNA/lncRNA pairs defined in Expanded View **Table EV3**.

Fig.2b,2c,4g,5h,7c,7d

When conducting ANOVA, it is mandatory to provide F-statistics. Two-way ANOVA should be used to examine the interaction of two factors, and the author should use the correct statistical methods or provide statements regarding such interactions.

> As requested, in **Figs. 2C, 2C, 4G, 5H, 7C, and 7D** we have now added F-statistics indicated as Interaction P.

Fig.3b, 4b, 5d, 5e, 5f, 5g

Why were the values in the Vehicle group the same? It's unlikely that all values are identical. A Student t-test is inappropriate here because the Vehicle group variance is 0.

> We would like to thank the reviewer for pointing out this error. First, the experiments displayed in panels **Figs. 3B, 4B, 5D, 5E, 5F, and 5G** are three individual biological replicates of three technical replicates, and as such the controls are set to '1'. The reviewer points out correctly that the Students T-test is inappropriate, and therefore we have adjusted the statistics to a one-sided T-test and Wilcoxon test.

Fig.5c is unnecessary because the cells used in this dataset are irrelevant to this manuscript.

> We would like to thank the reviewer for their feedback. We have removed the topology map from the main figure since it was not done in mouse macrophages. However, we believe that the data provide valuable insight and have moved the panel to the Expanded View appendix **Fig. EV3C**.

Fig.6c, 6d

The statistical significance in RNA-seq data should be determined using negative binomial-based statistics, such as DESeq2, rather than the Student t-test.

>> We would like to thank the reviewer for pointing out this mistake in the Figure Legends. DESeq2 was used to calculate the P-values in **Fig. 6C and 6D**. This has now been corrected.

December 24, 2024

RE: Life Science Alliance Manuscript #LSA-2024-03021-TR

Dr. Coen van Solingen
New York University
NYU Cardiovascular Research Center
435 E 30th Street
Science Building 704
New York, NY 10026

Dear Dr. van Solingen,

Thank you for submitting your revised manuscript entitled "LncRNA CARINH regulates expression and function of innate transcription factor IRF1 in macrophages". We would be happy to publish your paper in Life Science Alliance pending final revisions necessary to meet our formatting guidelines.

-please be sure that the authorship listing and order is correct
-we allow supplemental figures and tables, but not expanded view. Please label accordingly and update any callouts referring to expanded view figures or tables.

Figure Check:

-please add scale bars to Figure 4H

A. FINAL FILES:

B. MANUSCRIPT ORGANIZATION AND FORMATTING:

Sincerely,

Reviewer #3 (Comments to the Authors (Required)):

The manuscript titled " LncRNA IRF1-AS1 regulates expression and function of innate transcription factor IRF1 in human and mouse macrophages" by Cyr et al. demonstrated the impact of the IRF1-AS1 lncRNA gene, by mediating IRF1 gene, on viral infection.

The authors made a lot of effort on improving the manuscript. I think now the manuscript is easy to understand, and has enough evidences to claim that inducible Carinh upon interferon signaling stabilizes IRF1, leading to protect us from viral infection.

December 30, 2024

RE: Life Science Alliance Manuscript #LSA-2024-03021-TRR

Dr. Coen van Solingen
New York University
NYU Cardiovascular Research Center
435 E 30th Street
Science Building 704
New York, NY 10026

Dear Dr. van Solingen,

Thank you for submitting your Research Article entitled "LncRNA CARINH regulates expression and function of innate transcription factor IRF1 in macrophages". It is a pleasure to let you know that your manuscript is now accepted for publication in Life Science Alliance. Congratulations on this interesting work.

DISTRIBUTION OF MATERIALS:

Again, congratulations on a very nice paper. I hope you found the review process to be constructive and are pleased with how the manuscript was handled editorially. We look forward to future exciting submissions from your lab.

Sincerely,
